# Optimization Dynamics of Equivariant and Augmented Neural Networks

**Oskar Nordenfors**                                          *oskar.nordenfors@umu.se*
*Department of Mathematics and Mathematical Statistics*
*Umeå University*

**Fredrik Ohlsson**                                          *fredrik.ohlsson@umu.se*
*Department of Mathematics and Mathematical Statistics*
*Umeå University*

**Axel Flinth**                                          *axel.flinth@umu.se*
*Department of Mathematics and Mathematical Statistics*
*Umeå University*

**Reviewed on OpenReview:** *https://openreview.net/forum?id=PTTa3U29NR*

## Abstract

We investigate the optimization of neural networks on symmetric data, and compare the strategy of constraining the architecture to be equivariant to that of using data augmentation. Our analysis reveals that the relative geometry of the admissible and the equivariant layers, respectively, plays a key role. Under natural assumptions on the data, network, loss, and group of symmetries, we show that compatibility of the spaces of admissible layers and equivariant layers, in the sense that the corresponding orthogonal projections commute, implies that the sets of equivariant stationary points are identical for the two strategies. If the linear layers of the network also are given a unitary parametrization, the set of equivariant layers is even invariant under the gradient flow for augmented models. Our analysis however also reveals that even in the latter situation, stationary points may be unstable for augmented training although they are stable for the manifestly equivariant models.

## 1 Introduction

In machine learning, the general goal is to find 'hidden' patterns in data. However, there are sometimes symmetries in the data that are known a priori. Incorporating these manually should, heuristically, reduce the complexity of the learning task. In this paper, we are concerned with training neural networks on data exhibiting symmetries that can be formulated as equivariance under a group action. A standard example is the case of translation invariance in image classification.

More specifically, we want to theoretically study the connections between two general approaches to incorporating symmetries. The first approach is to construct equivariant models by means of *architectural design*. This framework, known as *Geometric Deep Learning* (Bronstein et al., 2017; 2021), exploits the geometric origin of the group $G$ of symmetry transformations by choosing the linear layers and nonlinearities to be equivariant (or invariant) with respect to the action of $G$. In other words, the symmetry transformations commute with each linear (or affine) map in the network, which results in an architecture which manifestly respects the symmetry $G$ of the problem. One prominent example is the spatial weight sharing of convolutional neural networks (CNNs) which are equivariant under translations. Group equivariant convolution networks (GCNNs) (Cohen and Welling, 2016; Weiler et al., 2018; Kondor and Trivedi, 2018) extends this principle to an exact equivariance under more general symmetry groups. The second approach is agnostic to model architecture, and instead attempts to achieve equivariance during training via *data augmentation*, which

refers to the process of extending the training to include synthetic samples obtained by subjecting training data to random symmetry transformations.

Both approaches have their benefits and drawbacks. Equivariant models use parameters efficiently through weight sharing along the orbits of the symmetry group, but are difficult to implement and computationally expensive to evaluate in general, since they entail numerical integration over the symmetry group (see, e.g., Kondor and Trivedi (2018)). Data augmentation, on the other hand, is agnostic to the model structure and easy to adapt to different symmetry groups. However, the augmentation strategy is by no means guaranteed to achieve a model which is exactly equivariant: the hope is that model will 'automatically' infer invariances from the data, but there are few theoretical guarantees. Also, augmentation in general entails an inefficient use of parameters and an increase in model complexity and training required.

In this paper, we study and compare the training dynamics of the two strategies as follows. We consider a nominal architecture (i.e. not equivariant by design) defined by restricting the linear layers of a multilayer perceptron (i.e. fully connected neural network without biases) to a certain affine subspace $\mathcal{L}$. In this way, we can treat many commonly used architectures, such as CNNs, transformer architectures, recurrent neural networks (RNNs), etc. We then train it using gradient flow, either on augmented data or while restricting the weights to also lie in the space of equivariant linear maps $\mathcal{H}_G$. Our results apply to all compact groups.

Our analysis reveals that under a few natural assumptions, including that the augmentation is performed with respect to the *Haar measure*, and a *compatibility assumption* (that the orthogonal projections onto $\mathrm{T}\mathcal{L}$ and onto $\mathcal{H}_G$ commute) a surprisingly simple relation between the sets of stationary points $S^{\mathrm{aug}}$ and $S^{\mathrm{eqv}}$ *lying in* $\mathcal{E} := \mathcal{L} \cap \mathcal{H}_G$ of the augmented model and the restricted model respectively:

(i) $S^{\mathrm{eqv}} = S^{\mathrm{aug}}$ (Theorem 3.7). In other words, augmentation neither introduces new equivariant stationary points, nor does it exclude existing ones, compared to restricting the architecture.

(ii) A stationary point in $\mathcal{E}$ can simultaneously be stable for the equivariant strategy while unstable for the the augmented one, but not vice-versa. (Theorem 3.12). In other words, while the equivariant and augmented models have the same stationary points in $\mathcal{E}$, some of them may be impossible to actually obtain during training for the augmented ones. In particular, it is not guaranteed that the augmented model has any local minima in $\mathcal{E}$.

We also show that under additional assumptions on the implementation of the architecture (related to how $\mathcal{L}$ is parametrized), $\mathcal{E}$ becomes an invariant set under the augmented flow (Theorem 3.7, part 2). Finally, we perform some simple numerical experiments to illustrate our findings.

## 1.1 Related work

The group theory based model for group augmentation we use here is heavily inspired by the framework developed in Chen et al. (2020). Augmentation and manifest invariance/equivariance have been studied from this perspective in a number of papers (Lyle et al., 2019; 2020; Mei et al., 2021; Elesedy and Zaidi, 2021). More general models for data augmentation have also been considered (Dao et al., 2019). Previous work has mostly been concerned with so-called kernel and *feature-averaged* models, and in particular, fully general neural networks as we treat them here have not been considered. The works have furthermore mostly been concerned with proving statistical properties of the models, and not with studying their dynamics at training. An exception is Lyle et al. (2020), in which it is proven that in linear scenarios, the equivariant models are optimal, but little is known about more involved models.

The dynamics of training *linear equivariant networks* (i.e., MLPs without nonlinearities) has been given some attention in the literature. Linear networks is a simplified, but nonetheless popular theoretical model for analysing neural networks (Bah et al., 2022). In Lawrence et al. (2022), the authors analyse the implicit bias of training a linear neural network with one fully connected layer on top of an equivariant backbone using gradient descent. They also provide some numerical results for non-linear models, but no comparison to data augmentation is made. In Chen and Zhu (2024), completely equivariant linear networks are considered, and an equivalence result between augmentation and restriction is proven for binary classification tasks. However, more realistic MLPs involving non-linearities are not treated at all.

Empirical comparisons of training equivariant and augmented non-equivariant models are common in the literature. Most often, the augmented models are considered as baselines for the evaluation of the equivariant models. More systemic investigations include Gandikota et al. (2021); Müller et al. (2021); Gerken et al. (2022). Compared to previous work, our formulation differs in that the parameter of the augmented and equivariant models are defined on the same vector spaces, which allows us to make a stringent mathematical comparison.

## 2 Mathematical framework

Let us begin by setting up the framework (see Figure 1). We let $X$ and $Y$ be vector spaces and $\mathcal{D}(x, y)$ be a joint distribution on $X \times Y$. We are concerned with training an neural network $\Phi_A : X \to Y$ so that $y \approx \Phi_A(x)$ is probable (with respect to $\mathcal{D}$). The network has the form

$$x_0 = x, \quad x_{i+1} = \sigma_i(A_i x_i), \quad i \in [L] = \{0, \ldots, L-1\}, \quad \Phi_A(x) = x_L, \tag{1}$$

where $A_i : X_i \to X_{i+1}$ are linear maps (layers) between (hidden) vector spaces $X_i$ with $X = X_0$ and $Y = X_L$, and $\sigma_i : X_{i+1} \to X_{i+1}$ are non-linearities. Note that $A = (A_i)_{i \in [L]}$ parametrizes the network since the non-linearities are assumed to be fixed. Let us denote the space of all possible parameters $\mathcal{H} = \oplus_{i \in [L]} \mathrm{Hom}(X_i, X_{i+1})$.

Note that while the network is well-defined for any choices of $A_i \in \mathrm{Hom}(X_i, X_{i+1})$, we may restrict the layer to some subset of $\mathrm{Hom}(X_i, X_{i+1})$ to define other architectures. Here, we assume that layers are confined to an *affine subspace* $\mathcal{L} \subseteq \mathcal{H}$. We will refer to the latter as the space of *admissible maps*. Note that this model, while simple, encompasses many popular architectures (fully connected layers with and without bias, residual layers, convolutional layers, recurrent networks and also attention layers). We explain this in detail in Appendix (A).

Two simple examples are fully connected layers without bias, and convolutional layers. In the former case, $\mathcal{L} = \mathcal{H}$, and in the latter, $\mathcal{L} = \bigoplus_{i \in [L]} \mathrm{C}(X_i, X_{i+1})$, where $C(X_i, X_{i+1})$ denotes the linear subspace of convolutional operators between $X_i$ and $X_{i+1}$. While simple, these examples encapsulate many aspects of the framework, and serve well as a guide for our development.

### 2.1 Representation theory and equivariance

Throughout the paper, we aim to make the neural network *equivariant* towards a group of symmetry transformations of the space $X \times Y$. That is, we consider a group $G$ acting on the vector spaces $X$ and $Y$ through *representations* $\rho_X$ and $\rho_Y$, respectively. A representation $\rho$ of a group $G$ on a vector space $V$ is a map from the group $G$ to the group of invertible linear maps $\mathrm{GL}(V)$ on $V$ that respects the group operation, i.e. $\rho(gh) = \rho(g)\rho(h)$ for all $g, h \in G$. The representation $\rho$ is unitary if $\rho(g)$ is unitary for all $g \in G$.

Given representations $\rho_U$ and $\rho_V$ on vector spaces $U$ and $V$ respectively, we may naturally define a lifted representation on $\mathrm{Hom}(U, V)$ as follows:

$$\bar{\rho}(g)M = \rho_V(g)M\rho_U(g)^{-1}, \ M \in \mathrm{Hom}(U, V) \tag{2}$$

It is easy to see that if $\rho_U$ and $\rho_V$ are unitary, so is $\bar{\rho}$.

In this paper, we are concerned with training *equivariant* models. A function $f : X \to Y$ is called equivariant with respect to $G$ if $f \circ \rho_X(g) = \rho_Y(g) \circ f$ for all $g \in G$ – that is, applying $f$ to a transformed example $\rho(g)x$ yields the same result as first applying $f$ and then transforming it with $\rho_Y(g)$.

For future reference, let us define the space of linear equivariant maps between $U$ and $V$ as $\mathrm{Hom}_G(U, V)$. Note that the lifted representation $\bar{\rho}$ is connected to equivariance of linear maps: We have $\bar{\rho}(g)M = M$ for all $g \in G$ if and only if $M \in \mathrm{Hom}_G(U, V)$.

We recall some important examples of representations.

*Example* 2.1. A simple, but important, representation is the trivial one, $\rho^{\mathrm{triv}}(g) = \mathrm{id}$ for all $g \in G$. If we equip $Y$ with the trivial representation, the equivariant functions $f : X \to Y$ are the invariant ones.

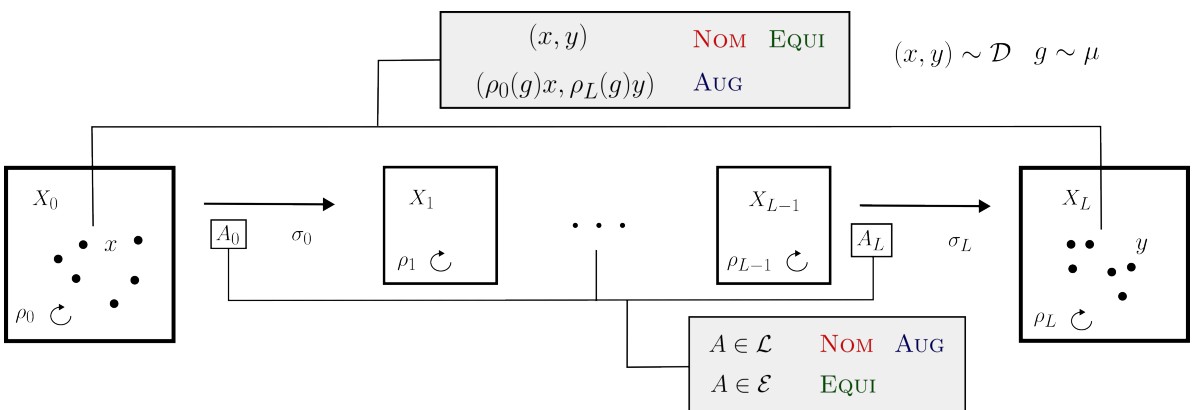

Figure 1: A graphical summary of our framework. The difference between the nominal network and the augmented one lies in the data, the difference between the nominal network and the equivariant one lies in restricting the layers.

*Example* 2.2. $\mathbb{Z}_N^2$ acts through translations on images $x \in \mathbb{R}^{N,N}$: $(\rho^{\text{trans}}(k,\ell)x)[i,j] = x[i-k,j-\ell]$.

*Example* 2.3. $\mathbb{Z}_4$ acts via discrete rotations of $\pi/2$ on images $x \in \mathbb{R}^{N,N}$: If $\omega : [N]^2 \to [N]^2$ describes the rotation of $\pi/2$ counter clockwise in pixel space, the representation is given by $(\rho^{\text{rot}}(k)x)[\ell] = x[\omega^k \ell]$, $\ell \in [N]^2$.

## 2.2 Two strategies for obtaining equivariant models

Let us now consider the task of training the model $\Phi_A$ to fit an equivariant target, i.e. an equivariant function $f : X \to Y$ for which the training data and labels fulfill $y = f(x)$. While possible, it is far from clear that a simple risk minimization

$$\min_{A \in \mathcal{L}} R(A) = \min_{A \in \mathcal{L}} \mathbb{E}_{\mathcal{D}}(\ell(\Phi_A(x), y)), \tag{3}$$

where $\ell : Y \times Y \to \mathbb{R}$ is some loss-function, yields an $A \in \mathcal{L}$ so that $\Phi_A$ is an equivariant function. That is, a minimization of the nominal risk (3) does not take advantage of the inductive bias of the equivariance of the ground truth $f$. We will in this paper analyze two strategies for doing so.

**Strategy 1: Manifest equivariance** The first method of enforcing equivariance is to constrain the layers to be manifestly equivariant. That is, we assume that $G$ is acting also on all hidden spaces $X_i$ through representations $\rho_i$, where $\rho_0 = \rho_X$ and $\rho_L = \rho_Y$, and constrain each layer $A_i$ to be equivariant: i.e., we choose $A_i$ to lie in the space $\text{Hom}_G(X_i, X_{i+1})$ of equivariant maps. Defining $\mathcal{H}_G = \bigoplus_{i \in [L]} \text{Hom}_G(X_i, X_{i+1})$, we hence constrain the $A \in \mathcal{L}$ to the *equivariant subspace*

$$\mathcal{E} = \mathcal{L} \cap \mathcal{H}_G. \tag{4}$$

If we in addition assume that all non-linearities $\sigma_i$ are equivariant, it is straight-forward to show that $\Phi_A$ is *exactly* equivariant under $G$ (see also Lemma 3.10). We will refer to these models as *equivariant*. A convenient way to formulate this strategy is to choose an $A$ which solves the following optimization problem

$$\min_{A \in \mathcal{E}} R(A). \tag{5}$$

The set $\mathcal{E}$, or rather $\mathcal{H}_G$, has been extensively studied in the setting of geometric deep learning and explicitly characterized in many important cases (Maron et al., 2019a; Cohen et al., 2019; Kondor and Trivedi, 2018; Weiler and Cesa, 2019; Maron et al., 2019b; Aronsson, 2022). In Finzi et al. (2021), a general method for determining $\mathcal{H}_G$ numerically directly from the $\rho_i$ and the structure of the group $G$ is described.

| | Def. | Clarification | Projection |
|---|---|---|---|
| $\mathcal{H}$ | $\bigoplus_{i \in [L]} \mathrm{Hom}(X_i, X_{i+1})$ | All possible layers | – |
| $\mathcal{L}$ | – | Admissible layers | $\Pi_{\mathcal{L}}$ |
| $\mathcal{H}_G$ | $\bigoplus_{i \in [L]} \mathrm{Hom}_G(X_i, X_{i+1})$ | All equivariant layers | $\Pi_G$ |
| $\mathcal{E}$ | $\mathcal{L} \cap \mathcal{H}_G$ | Admissible equiv. layers | $\Pi_{\mathcal{E}}$ |

Table 1: Four important spaces. Note that the orthogonal projections technically are onto the tangent spaces $T\mathcal{L}$ and $T\mathcal{E}$.

**Strategy 2: Data augmentation**  The second method we consider is to augment the training data. To this end, we define a new distribution on $X \times Y$ by drawing samples $(x, y)$ from $\mathcal{D}$ and subsequently *augmenting* them by applying the action of a randomly drawn group element $g \in G$ on both data $x$ and label $y$. Training on this augmented distribution can be formulated as optimizing the *augmented risk*

$$R^{\mathrm{aug}}(A) = \int_G \mathbb{E}_{\mathcal{D}}(\ell(\Phi_A(\rho_X(g)x), \rho_Y(g)y)) \, \mathrm{d}\mu(g) \tag{6}$$

Here, $\mu$ is the normalised *Haar* measure on the group, which is defined through its invariance with respect to the action of $G$ on itself; if $h$ is distributed according to $\mu$ then so is $gh$ for all $g \in G$. This property of the Haar measure will be crucial in our analysis. Choosing another measure would cause the augmentation to be biased towards certain group elements, and is not considered here. The normalised Haar measure exists since we have assumed that $G$ is compact (Krantz and Parks, 2008). Note that if the data $\mathcal{D}$ already is symmetric in the sense that $(x, y) \sim (\rho_X(g)x, \rho_Y(g)y)$, the augmentation acts trivially.

*Remark* 2.4. Equation (6) is a simplification – in practice, the actual function that is optimized is an empirical approximation of $R^{\mathrm{aug}}$ formed by sampling of the group $G$. Our results are hence about an 'infinite-augmentation limit' that still should have high relevance at least in the 'high-augmentation region' due to the law of large numbers. To properly analyse this transition carefully is important, but beyond the scope of this work.

## 3  The dynamics of gradient flow near $\mathcal{E}$

As has been outlined in the previous sections, the three training strategies can be formulated as three differing optimization problems:

$$\textsc{Nom}: \min_{A \in \mathcal{L}} R(A) \qquad \textsc{Aug}: \min_{A \in \mathcal{L}} R^{\mathrm{aug}}(A) \qquad \textsc{Equi}: \min_{A \in \mathcal{E}} R(A).$$

We will study *parametrized gradient flows* corresponding to these optimization problems. Concretely, let us write $\mathcal{L} = A_0 + T\mathcal{L}$ and $\mathcal{E} = A_0 + T\mathcal{E}$ for some common 'base-point' $A_0 \in \mathcal{E}$ and respective tangent spaces $T\mathcal{L}$ and $T\mathcal{E}$. Letting $L : \mathbb{R}^{\dim \mathcal{L}} \to \mathcal{H}$ and $E : \mathbb{R}^{\dim \mathcal{E}} \to \mathcal{H}$ be embedding operators with $\mathrm{im}(L) = T\mathcal{L}$ and $\mathrm{im}(E) = T\mathcal{E}$, we solve the optimization problems via applying gradient flow to the functions

$$\mathrm{R}^{\mathrm{nom}}(c) := R(A_0 + Lc), \quad \mathrm{R}^{\mathrm{aug}}(c) := R^{\mathrm{aug}}(A_0 + Lc), \quad \mathrm{R}^{\mathrm{eqv}}(c) := R(A_0 + Ec), \tag{7}$$

respectively. This strategy is by far the most common in practice, and entails evolving the coefficient vectors according to $\dot{c} = -\nabla \mathrm{R}^{\mathrm{nom}}(c)$, and so forth. By applying the chain rule, we furthermore see that this causes the following evolutions of the $A$:

$$\textsc{Nom}: \dot{A} = -LL^*\nabla R(A), \quad \textsc{Aug}: \dot{A} = -LL^*\nabla R^{\mathrm{aug}}(A), \quad \textsc{Equi}: \dot{A} = -EE^*\nabla R(A). \tag{8}$$

To go a bit more into detail, consider for instance the $\textsc{Nom}$ dynamics. First, we have $\dot{A} = L\dot{c}$ and $\dot{c} = -\nabla \mathrm{R}^{\mathrm{nom}}(c)$. The chain rule furthermore implies $\nabla \mathrm{R}^{\mathrm{nom}}(c) = L^*\nabla R(A_0 + Lc) = L^*\nabla R(A)$, which yields the formula.

*Remark* 3.1. A special case is the embedding operators $L$, or $E$, being 'partially unitary' (in the sense that $L^*L = \text{id}$, or $E^*E = \text{id}$). In this case, $LL^*$ equals the orthogonal projection $\Pi_{\mathcal{L}}$ onto T$\mathcal{L}$, and likewise, $EE^*$ equals the orthogonal projection $\Pi_{\mathcal{E}}$ onto T$\mathcal{E}$. Consequently, $A$ evolves according to projected gradient flows in this case.

To conduct our analysis, we need three global assumptions.

**Assumption 1.** The group $G$ is compact, and is acting on all hidden spaces $X_i$ through unitary representations $\rho_i$. The spaces $\text{Hom}_G(X_i, X_{i+1})$ are all non-empty.

**Assumption 2.** The non-linearities $\sigma_i : X_{i+1} \to X_{i+1}$ are equivariant.

**Assumption 3.** The loss $\ell$ is invariant, i.e. $\ell(\rho_Y(g)y, \rho_Y(g)y') = \ell(y, y')$, $y, y' \in Y$, $g \in G$.

Let us briefly comment on these assumptions. First, the compactness assumption is needed to ensure the existence of the normalised Haar measure. While this includes all finite groups, and the orthogonal groups $\text{SO}(n)$ and $\text{O}(n)$, it should be noted that it excludes for example the group of all rigid motions $\text{SE}(n)$. The non-emptyness assumption is needed for the restriction strategy to be well defined. The assumption of unitarity, i.e., that $\rho_i(g)$ preserves the inner product $\langle \rho_i(g)x'_i, \rho_i(g)x_i \rangle = \langle x'_i, x_i \rangle$ for all $g \in G$ and $x_i, x'_i \in X_i$, is not a true restriction: As long as all $X_i$ are finite-dimensional, we can (since $G$ is compact) redefine the inner products on $X_i$ to ensure that all $\rho_i$ become unitary. The second assumption is required for the equivariant strategy to be sound – if the $\sigma_i$ are not equivariant, they will explicitly break equivariance of $\Phi_A$ even if $A \in \mathcal{E}$. The third assumption guarantees that the loss-landscape is 'unbiased' towards group transformations, which is certainly required to train any model respecting the symmetry group.

We also note that all assumptions are in many settings quite weak. We already commented on Assumption 1. As for Assumption 2, note that any non-linearity acting pixel-wise on an image will be equivariant to any representation acting by moving around the pixels of the image, for instance translations and rotations. In the same way, any loss comparing images pixel by pixel will satisfy Assumption 3. This is not to say that there are not cases where point-wise non-linearities fail to satisfy the equivariance assumptions. For example, when dealing with point-cloud data, or steerable networks incorporating transformations under general representations of $G$, not any 'point-wise' non-linearity is equivariant, and more care needs to be taken. We refer to, e.g. (Gerken et al., 2022; Deng et al., 2021), for a more in-depth discussion and examples of constructions of equivariant non-linearities in these cases. Let us finally point out that if we are trying to learn an invariant function the final representation $\rho_Y$ is trivial and Assumption 3 is trivially satisfied. We discuss more cases in the Appendix A.

## 3.1 The compatibility condition

We now come to the analysis of the stationary points, lying in $\mathcal{E}$, of the three dynamics in Equation 8. Our main result will be that a geometric condition on the relation of $\mathcal{L}$ and $\mathcal{E}$ will imply that the stationary points of AUG and EQUI on the subspace $\mathcal{E}$ are exactly the same. Let us formulate the condition.

**Definition 3.2.** We say that the *compatibility condition* is satisfied for a space of admissible maps $\mathcal{L}$ if $\Pi_{\mathcal{L}}$ commutes with the orthogonal projection $\Pi_G$ onto $\mathcal{H}_G$.

The compatibility condition is equivalent to the following arguably more useful statement.

**Lemma 3.3.** *The compatibility condition is equivalent to* $\Pi_{\mathcal{L}}\Pi_G = \Pi_{\mathcal{E}}$.

The simple proof is given in Appendix D. A common case when the compatibility condition holds is when $\mathcal{L}$ is invariant to the lifted representation (2) of $G$ on $\mathcal{H}$ (defined layerwise $(\overline{\rho}(g)A)_i = \overline{\rho}(g)A_i$).

**Proposition 3.4.** *If $\mathcal{L}$ is invariant under all $\overline{\rho}(g)$, $g \in G$, the compatibility condition 3.2 is satisfied.*

To prove both Proposition 3.4 and the main result, we will need the following well-known relation, which can be found in e.g. Fulton and Harris (2004, Prop. 2.8), sometimes referred to as the *twirling formula*. For the convenience of the reader, we provide a proof in Appendix D.

**Lemma 3.5.** (Twirling formula) *Letting $\mu$ denote the Haar measure of the group $G$, we have*

$$\Pi_G A = \int_G \overline{\rho}(g) A \, \mathrm{d}\mu(g).$$

Let us now prove Proposition 3.4.

*Proof of Proposition 3.4.* Suppose that $\overline{\rho}(g)$ leaves $\mathcal{L}$ invariant. Then it also leaves $\mathrm{T}\mathcal{L}$ invariant. Thus, for any $A \in \mathrm{T}\mathcal{L}$ and $B \in \mathrm{T}\mathcal{L}^{\perp}$ we have

$$\langle A, \overline{\rho}(g)B \rangle = \langle \overline{\rho}(g)^{-1}A, B \rangle = \langle \overline{\rho}(g^{-1})A, B \rangle = 0$$

by unitarity of representations. That is to say, each $\overline{\rho}(g)$ leaves the orthogonal complement $\mathrm{T}\mathcal{L}^{\perp}$ invariant, so that each $\overline{\rho}(g)$ commutes with $\Pi_{\mathcal{L}}$. By Lemma 3.5, $\Pi_G$ hence commutes with $\Pi_{\mathcal{L}}$. $\qquad\square$

For most canonical examples, $\mathcal{L}$ is indeed invariant under $\overline{\rho}$. For instance, for the fully connected architectures without bias terms, $\mathcal{L} = \mathcal{H}$, so the invariance holds trivially. In the A ppendix B, we show that $\mathcal{L}$ is invariant under $\overline{\rho}$ for many other architectures and group actions as well. We also show that Proposition 3.4 does not give a necessary condition for the compatibility condition to hold. Let us here limit the discussion to one interesting case.

*Example* 3.6. Consider $U = V = \mathbb{R}^{N,N}$, the discrete rotation action $\rho^{\mathrm{rot}}$ of $\mathbb{Z}_4$ on it (let us write $\rho$ instead of $\rho^{\mathrm{rot}}$ to simplify notation), and the set of convolutional operators

$$\mathcal{C} = \left\{ C_\varphi \in \mathrm{Hom}(U, V) \,\middle|\, (C_\varphi x)[\ell] = \sum_{j \in [N]^2} x[\ell - j]\varphi[j], \, \ell \in [N]^2 \text{ for a } \varphi : [N]^2 \to \mathbb{R} \right\}$$

as the nominal architecture. The lifted $\overline{\rho}$ is acting directly on the filter: $\overline{\rho}(k)C_\varphi = C_{\rho(k)\varphi}$. To show that, we need to show that $\rho(k)C_\varphi \rho(k)^{-1}x = C_{\rho(k)\varphi}x$ for all $x \in U, k \in \mathbb{Z}_4$. We have

$$(C_{\rho(k)\varphi}x)[\ell] = \sum_{j \in [N]^2} x[\ell - j](\rho(k)\varphi)[j] = \sum_{j \in [N]^2} x[\ell - j]\varphi[\omega^k j] = \sum_{j \in [N]^2} x[\ell - \omega^{-k}j]\varphi[j]$$

$$= \sum_{j \in [N]^2} (\rho(k)^{-1}x)[\omega^k \ell - j]\varphi[j] = (C_\varphi \rho(k)^{-1}x)[\omega^k \ell] = (\rho(k)C_\varphi \rho(k)^{-1}x)[\ell].$$

This immediately shows that $\overline{\rho}(k)\mathcal{C} \subseteq \mathcal{C}$ for all $k$. However, in practice, one might also want to restrict the filters, e.g. by restricting its support to a set $\Omega \subseteq [N]^2$, i.e. to

$$\mathcal{L}_\Omega = \{ C_\varphi \in \mathcal{C} \mid \mathrm{supp}\, \varphi \subseteq \Omega \}$$

A common choice is to restrict the filters to only be $3 \times 3$ or $5 \times 5$ pixels. For odd $N$ this corresponds to letting $\Omega$ equal a small square in the center of the image (note that $N$ can always be made odd through zero-padding). For such rotation symmetric $\Omega$, the space $\mathcal{L}_\Omega$ again becomes invariant under all $\overline{\rho}(k)$.

We could however consider other choices for $\Omega$, such as in Figure 2. For a pattern like the left, cross-shaped one, $\overline{\rho}(k)\mathcal{L}_\Omega \subseteq \mathcal{L}_\Omega$ holds, since rotating a cross-shaped filter yields another cross-shaped filter. Thus, for convolution operators given by such a filter, the compatibility condition holds by Lemma 3.4. For non-symmetric $\Omega$ such as the right one, we do not have $\overline{\rho}(k)\mathcal{L}_\Omega \subseteq \mathcal{L}_\Omega$. In fact, one can even show that the compatibility condition 3.2 is not satisfied for $\mathcal{L}_\Omega$ in that case. We postpone these somewhat involved calculations to Appendix B – and return to the example in the numerical experiments in Section 4.

## 3.2 Stationary points

We can now formulate and prove the main result of the paper.

**Theorem 3.7.** *Under the compatibility condition 3.2, we have*

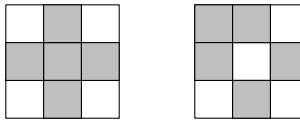

Figure 2: On the left: filter with cross-shaped support. On the right: filter with skew support. Grey indices correspond to non-zero indices. White indices correspond to zeroed-out indices.

1. *The sets of stationary points on $\mathcal{E}$, $S^{\mathrm{nom}}$, $S^{\mathrm{eqv}}$, and $S^{\mathrm{aug}}$, of NOM, EQUI and AUG, respectively, i.e.*

$$S^{\mathrm{nom}} := \{A \in \mathcal{E} \mid LL^*\nabla R(A) = 0\}, \quad S^{\mathrm{eqv}} := \{A \in \mathcal{E} \mid EE^*\nabla R(A) = 0\},$$
$$\text{and} \quad S^{\mathrm{aug}} := \{A \in \mathcal{E} \mid LL^*\nabla R^{\mathrm{aug}}(A) = 0\},$$

*satisfy $S^{\mathrm{nom}} \subseteq S^{\mathrm{eqv}} = S^{\mathrm{aug}}$.*

2. *If the embedding operators $L$ and $E$ are partially unitary, then $\mathcal{E}$ is an invariant set of the gradient flow of AUG.*

*Remark* 3.8. Note that Theorem 3.7 only applies to *points in $\mathcal{E}$*. It does not say anything about stationarity of points that are not in $\mathcal{E}$.

*Remark* 3.9. Although we have formulated our theory for gradient flow, Theorem 3.7 directly extends to (non-stochastic) gradient descent – a point is stationary for the gradient flow if and only if it is stationary for gradient descent. Since $\mathcal{E}$ is a vector space (and not a more general manifold), the same argument goes through for the second part. The extension to SGD is more complicated, and deemed outside the scope of this work.

The key to prove Theorem 3.7 is the following auxiliary result

**Lemma 3.10.** *The augmented risk can be expressed as*

$$R^{\mathrm{aug}}(A) = \int_G R(\overline{\rho}(g)A) \, \mathrm{d}\mu(g). \tag{9}$$

*Consequently for $A \in \mathcal{E}$*

$$\nabla R^{\mathrm{aug}}(A) = \Pi_G \nabla R(A).$$

*Proof.* It is enough to show that for $A \in \mathcal{L}$ and $g \in G$, we have

$$\Phi_A(\rho_X(g)x) = \rho_Y(g)\Phi_{\overline{\rho}(g)^{-1}A}(x). \tag{10}$$

Once that has been proven, the statement follows from Assumption 3:

$$R^{\mathrm{aug}}(A) = \int_G \mathbb{E}_{\mathcal{D}}(\ell(\Phi_A(\rho_X(g)x), \rho_Y(g)y))\mathrm{d}\mu \overset{(10)}{=} \int_G \mathbb{E}_{\mathcal{D}}(\ell(\rho_Y(g)\Phi_{\overline{\rho}(g)^{-1}A}(x), \rho_Y(g)y))\mathrm{d}\mu$$

$$\overset{\text{Ass. 3}}{=} \int_G \mathbb{E}_{\mathcal{D}}(\ell(\Phi_{\overline{\rho}(g)^{-1}A}(x), y))\mathrm{d}\mu = \int_G R(\overline{\rho}(g)^{-1}A) \, \mathrm{d}\mu(g) = \int_G R(\overline{\rho}(g)A) \, \mathrm{d}\mu(g),$$

where the final step is a property of the Haar measure: if $g \sim \mu$, then $g^{-1} \sim \mu$.

We proceed with the proof of (10). Using the notation from (1): $x_i$ denotes the output of layer $i$ of the network $\Phi_A$ when it acts on the input $x \in X$. Also, for $g \in G$, let $x_i^g$ denote the outputs of each layer of the network $\Phi_{\overline{\rho}(g)^{-1}A}$ when acting on the input $\rho_X(g)^{-1}x$. If we can show that

$$\rho_i(g)x_i^g = x_i, \quad i \in [L+1], \tag{11}$$

then the claim follows: For $i = L$, it reads $\rho_L(g)x_L^g = x_L$, which actually means $\rho_Y(g)\Phi_{\overline{\rho}(g)^{-1}A}(\rho_X(g)^{-1}x) = \Phi_A(x)$, which clearly is equivalent to (10).

We show (11) via induction. The case $i = 0$ is clear: $\rho_X(g)x_0^g = \rho_X(g)\rho_X(g)^{-1}x = x = x_0$. As for the induction step, we have

$$\rho_{i+1}(g)x_{i+1}^g = \rho_{i+1}(g)\sigma_i\big((\overline{\rho}(g)^{-1}A_i)x_i^g\big) \stackrel{\text{Def. } \overline{\rho}}{=} \rho_{i+1}(g)\sigma_i(\rho_{i+1}(g)^{-1}A_i\rho_i(g)x_i^g)$$

$$\stackrel{\text{Ass. } 2}{=} \sigma_i(\rho_{i+1}(g)\rho_{i+1}(g)^{-1}A_i\rho_i(g)x_i^g) = \sigma_i(A_i\rho_i(g)x_i^g) \stackrel{\text{Ind. ass.}}{=} \sigma_i(A_ix_i) = x_{i+1} \, ,$$

and the claim follows.

Now differentiate (9) with respect to $A$ to yield the the equality

$$\nabla R^{\text{aug}}(A) = \int_G \overline{\rho}(g)^*\nabla R(\overline{\rho}(g)A)\mathrm{d}\mu(g).$$

If $A \in \mathcal{E}$, $\overline{\rho}(g)A = A$ for all $g$. This, and the twirling formula (Lemma 3.5) implies, by unitarity of $\overline{\rho}$ and a property of the Haar measure, that the above equals $\Pi_G\nabla R(A)$, which was the claim. $\qquad\square$

Now we have all the tools necessary to prove the main result.

*Proof of Theorem 3.7.* Part 1: Equation (8) together with Lemma 3.10 tells us that stationary points of Nom, Aug and Equi on $\mathcal{E}$, respectively, are characterized by

$$LL^*\nabla R(A) = 0, \qquad LL^*\nabla R^{\text{aug}}(A) = LL^*\Pi_G\nabla R(A) = 0 \qquad EE^*\nabla R(A) = 0$$

We will show that under the compability condition, (a) $S^{\text{nom}} \subseteq S^{\text{eqv}}$, (b) $S^{\text{eqv}} \subseteq S^{\text{aug}}$ and (c) $S^{\text{aug}} \subseteq S^{\text{eqv}}$, which implies the statement.

(a) If $LL^*\nabla R(A) = 0$, then by linearity and injectivity of $L$ we have $L^*\nabla R(A) = 0$. In other words, we have $\nabla R(A) \in \ker L^* = (\text{im } L)^\perp = \mathrm{T}\mathcal{L}^\perp$. Now notice that $\mathrm{T}\mathcal{E} \subseteq \mathrm{T}\mathcal{L}$ by definition, which implies that $\mathrm{T}\mathcal{L}^\perp \subseteq \mathrm{T}\mathcal{E}^\perp$. Hence, $\nabla R(A)$ is also in $\mathrm{T}\mathcal{E}^\perp$, implying $E^*\nabla R(A) = 0$, and therefore also $EE^*\nabla R(A) = 0$. Thus, $S^{\text{nom}} \subseteq S^{\text{eqv}}$.

(b) If $EE^*\nabla R(A) = 0$, then by similar arguments as in (a) we have $\Pi_{\mathcal{E}}\nabla R(A) = 0$. By Lemma 3.3, $\Pi_{\mathcal{L}}\Pi_G\nabla R(A) = 0$, i.e. $\Pi_G\nabla R(A) \in \mathrm{T}\mathcal{L}^\perp = \ker L^*$, so that $LL^*\Pi_G\nabla R(A) = 0$. Thus, $S^{\text{eqv}} \subseteq S^{\text{aug}}$.

(c) If $LL^*\Pi_G\nabla R(A) = 0$, then by again similar arguments as in (a) we have $\Pi_{\mathcal{L}}\Pi_G\nabla R(A) = 0$. By Lemma 3.3, $\Pi_{\mathcal{E}}\nabla R(A) = 0$, i.e. $\nabla R(A) \in \mathrm{T}\mathcal{E}^\perp = \ker E^*$, so that $EE^*\nabla R(A) = 0$. Thus, $S^{\text{aug}} \subseteq S^{\text{eqv}}$.

Part 2: When $L$ is partially unitary, as we saw in Remark 3.1, we have $LL^* = \Pi_{\mathcal{L}}$, and consequently $\nabla R^{\text{aug}}(A) = \Pi_{\mathcal{L}}\Pi_G\nabla R(A)$ for $A \in \mathcal{E}$, which by Lemma 3.3 equals $\Pi_{\mathcal{E}}\nabla R(A)$ under the compatibility condition. Hence, the gradients at points in $\mathcal{E}$ lie in $\mathrm{T}\mathcal{E}$, which implies that $\mathcal{E}$ is an invariant set of the gradient flow of Aug. $\qquad\square$

## 3.3 Stability

Let us now consider the stability of stationary points in $\mathcal{E}$ for the three gradient flows. Note that when we say that a point $A \in \mathcal{E}$ is stable for Nom, what we really mean is that the $c$ so that $A = A_0 + Lc$ is a stable point for the dynamics $\dot{c} = -\nabla \mathrm{R}^{\text{nom}}(c)$, and similarly for Aug and Equi.

It is well known that as long as the Hessian of a function $f$ has no zero-eigenvalues, it can be used to classify the stability of a point $x_0$ under the gradient flow $\dot{x} = -\nabla f(x)$: $x_0$ is a stable point, more specifically a strict local minimum of $f$, if and only if $f''(x_0)$ is positive definite. Consequently, we will study the Hessians of $\mathrm{R}^{\text{nom}}$, $\mathrm{R}^{\text{aug}}$ and $\mathrm{R}^{\text{eqv}}$. We begin with the following statement about the Hessian of $R^{\text{aug}}$ in points $A \in \mathcal{E}$.

**Lemma 3.11.** *Let $A \in \mathcal{E}$. If we decompose $V \in \mathcal{H}$ as $X + Y$ with $X \in \mathcal{H}_G$ and $Y \in \mathcal{H}_G^\perp$, we have*

$$(R^{\text{aug}})''(A)[V, V] = R''(A)[X, X] + \int_G R''(A)[\overline{\rho}(g)Y, \overline{\rho}(g)Y]\mathrm{d}\mu(g).$$

*Proof.* Applying the chain rule to (10) yields

$$(R^{\mathrm{aug}})''(A)[V,V] = \int_G R''(\overline{\rho}(g)A)[\overline{\rho}(g)V,\overline{\rho}(g)V]\mathrm{d}\mu(g) = \int_G R''(A)[\overline{\rho}(g)V,\overline{\rho}(g)V]\mathrm{d}\mu(g),$$

since $\overline{\rho}(g)A = A$ for $A \in \mathcal{E}$. We now expand $R''(A)[\overline{\rho}(g)(X+Y),\overline{\rho}(g)(X+Y)]$. Since $X \in \mathcal{H}_G$, $\overline{\rho}(g)X = X$, and we obtain

$$\int_G \Big(R''(A)[X,X] + R''(A)[X,\overline{\rho}(g)Y] + R''(A)[\overline{\rho}(g)Y,X] + R''(A)[\overline{\rho}(g)Y,\overline{\rho}(g)Y]\Big)\mathrm{d}\mu(g)$$

$$= R''(A)[X,X] + R''(A)[X,\Pi_G Y] + R''(A)[\Pi_G Y,X] + \int_G R''(A)[\overline{\rho}(g)Y,\overline{\rho}(g)Y]\mathrm{d}\mu(g),$$

where we applied the twirling formula $\int_G \overline{\rho}(g)Y\,\mathrm{d}\mu(g) = \Pi_G Y$ and linearity. It is now only left to note that $\Pi_G Y = 0$, since $Y \in \mathcal{H}_G^\perp$. □

We can now analyse the stability of our flows.

**Theorem 3.12.** *Under the compatibility condition, we have*

    *1. If $A \in \mathcal{E}$ is a strictly stable point for NOM, it is also strictly stable for AUG and EQUI.*

    *2. If $A \in \mathcal{E}$ is a strictly stable point for AUG, it is also strictly stable for EQUI*

*Proof.* In the following, let $A$ denote a point in $\mathcal{E}$, $c$ the coefficient vector so that $A = A_0 + Lc$, and $e$ the one so that $A = A_0 + Ee$. Note that such coefficient vectors exist due to $A \in \mathcal{E}$.

Part 1: First, if $A \in \mathcal{E}$ is a stable point for NOM, it surely is stationary, and therefore by Theorem 3.7 a stationary point also for AUG and EQUI. Also, we must have $(R^{\mathrm{nom}})''(c)[d,d] > 0$ for all $d \neq 0$. The chain rule reveals that the latter means that $R''(A)[Ld,Ld] > 0$ for all $d \neq 0$, or equivalently $R''(A)[V,V] > 0$ for all $0 \neq V \in \mathrm{T}\mathcal{L}$. This in particular implies that

$$(R^{\mathrm{eqv}})''(e)[f,f] = R''(A)[Ef,Ef] > 0$$

for all $f \neq 0$, since $0 \neq Ef \in \mathrm{T}\mathcal{E} \subseteq \mathrm{T}\mathcal{L}$, so that $A$ is stable also for EQUI. As for AUG, we can for each $d \neq 0$ write $Ld = X + Y$ for $X \in \mathcal{H}_G$ and $Y \in \mathcal{H}_G^\perp$, with not both $X$ and $Y$ equal to 0, and by Lemma 3.11 obtain

$$(R^{\mathrm{aug}})''(c)[d,d] = (R^{\mathrm{aug}})''(A)[Ld,Ld] = R''(A)[X,X] + \int_G R''(A)[\overline{\rho}(g)Y,\overline{\rho}(g)Y]\mathrm{d}\mu(g) > 0,$$

so that $A$ is stable also for AUG. The strict inequality follows from the fact that if $X = 0$, we must have $Y \neq 0$, and therefore $\overline{\rho}(g)Y \neq 0$ for all $g \in G$.

Part 2: Suppose that $A \in \mathcal{E}$ is a stable point for AUG. We want to show that $(R^{\mathrm{eqv}})''(e)[f,f] = R''(A)[Ef,Ef] > 0$ for all $f \neq 0$. $Ef$ can, as a vector in $\mathrm{T}\mathcal{E} \subseteq \mathrm{T}\mathcal{L}$ we written as $Ld$ for some $d \neq 0$. In the decomposition $Ld = X + Y$ as in Lemma 3.11, $Y$ must be zero, since $Ld \in \mathrm{T}\mathcal{E} \subseteq \mathcal{H}_G$. Lemma 3.11 therefore implies that

$$(R^{\mathrm{aug}})''(c)[d,d] = (R^{\mathrm{aug}})''(A)[Ld,Ld] = R''(A)[Ld,Ld] = R''(A)[Ef,Ef].$$

Since $A$ is stable for AUG, we have $R''(A)[Ef,Ef] = (R^{\mathrm{aug}})''(c)[d,d] > 0$, which completes the proof. □

Theorem 3.12 gives a more nuanced meaning to Theorem 3.7. Although we have an equality of stationary points $S^{\mathrm{eqv}} = S^{\mathrm{aug}}$, it may very well be that a point in $\mathcal{E}$ is stable for EQUI but not for AUG. The contrary is however not possible. In this sense, using the EQUI flow is preferable if one searches for stationary points in $\mathcal{E}$ – they will then more often be stable. In the Appendix C, we explicitly construct examples with points that are stable for EQUI but not for AUG.

*Remark* 3.13. Let us quickly comment on the extension of Theorem 3.12 to gradient descent and SGD. As for gradient descent, we note that a strictly stable point for the gradient flow is also stable for gradient descent *if the learning rate is small enough.* Any quantitative statement would require assumptions on the Hessian of $R$ at $A$ (e.g. lower bounds on (restricted) eigenvalues), which would be impossible to check a priori. Again, the extension of the theorem to a stochastic setting is more involved, and will probably also depend on the 'degree of randomness' (e.g. batch sizes), and we leave it to future work.

Let us end this section by, utilizing similar ideas as above, derive an interesting statement about decoupling of the dynamics in $\mathcal{E}$ and in $\mathrm{T}\mathcal{E}^\perp$ which holds when $L$ and $E$ are partially unitary and the compatibility condition holds.

**Proposition 3.14.** *For $A \in \mathcal{L}$, write $A = X + Y$ with $X \in \mathcal{E}$ and $Y \in \mathrm{T}\mathcal{E}^\perp$. Assuming that the compatibility condition holds, and assuming that the embedding operator $L$ is partially unitary, the gradient flow of $R^{\mathrm{aug}}$ decouples in the following sense:*

$$\begin{cases} \dot{X} &= -\Pi_\mathcal{E} \nabla R(X) \qquad\quad + \mathcal{O}(\|Y\|^2) \\ \dot{Y} &= -\Pi_\mathcal{L}(R^{\mathrm{aug}})''(X)Y + \mathcal{O}(\|Y\|^2). \end{cases}$$

*In particular, when also $E$ is partially unitary, $X$ follows the* EQUI *dynamics up to $\mathcal{O}(\|Y\|^2)$.*

*Proof.* We have already argued that in this case, the dynamics are given by are given by $\dot{A} = -\Pi_\mathcal{L} \nabla R^{\mathrm{aug}}(A)$. Performing a Taylor expansion of $\nabla R^{\mathrm{aug}}(A)$ in $X$ yields

$$\Pi_\mathcal{L} \nabla R^{\mathrm{aug}}(A) = \Pi_\mathcal{L} \nabla R^{\mathrm{aug}}(X) + \Pi_\mathcal{L}(R^{\mathrm{aug}})''(X)Y + \mathcal{O}(|Y|^2).$$

Since $X \in \mathcal{E}$ and the compatibility condition holds, we have $\Pi_\mathcal{L} \nabla R^{\mathrm{aug}}(X) = \Pi_\mathcal{L} \Pi_G \nabla R(X) = \Pi_\mathcal{E} \nabla R(X)$.

Now, to determine the dynamics of $X$, let us test the equation $\dot{A} = -\Pi_\mathcal{L} \nabla R^{\mathrm{aug}}(A)$ with an arbitrary $W \in \mathrm{T}\mathcal{E}$. We namely have $\langle \dot{A}, W \rangle = \langle \dot{X}, W \rangle$ for such $W$. Furthermore, $\langle \Pi_\mathcal{L} \nabla R^{\mathrm{aug}}(X), W \rangle = \langle \Pi_\mathcal{E} \nabla R(X), W \rangle = \langle \nabla R(X), W \rangle$, and

$$\langle \Pi_\mathcal{L}(R^{\mathrm{aug}})''(X)Y, W \rangle = \langle (R^{\mathrm{aug}})''(X)Y, \Pi_\mathcal{L}W \rangle = (R^{\mathrm{aug}})''(X)[Y, \Pi_\mathcal{L}W].$$

Since $W \in \mathrm{T}\mathcal{E} \subseteq \mathrm{T}\mathcal{L}$, we have $\Pi_\mathcal{L}W = W$. Now, similarly as in the proof of Lemma 3.11, we apply the chain rule to $R^{\mathrm{aug}}$ and the fact that $X \in \mathcal{E} \subseteq \mathcal{H}_G$ to derive that

$$(R^{\mathrm{aug}})''(X)[Y, W] = \int_G R''(X)[\overline{\rho}(g)Y, \overline{\rho}(g)W] \, \mathrm{d}\mu(g) = R''(X)[\Pi_G Y, W].$$

where we used that $W \in \mathrm{T}\mathcal{E} \subseteq \mathcal{H}_G$, and also the twirling formula. Now, since $A = X + Y$, $Y$ must be in $\mathrm{T}\mathcal{L}$, and consequently $Y = \Pi_\mathcal{L}Y$. Consequently, appealing to the compatibility condition, $\Pi_G Y = \Pi_G \Pi_\mathcal{L}Y = \Pi_\mathcal{E}Y = 0$, since $Y \in \mathrm{T}\mathcal{E}^\perp$. Thus, $(R^{\mathrm{aug}})''(X)[Y, W] = 0$, and we get

$$\langle \dot{X}, W \rangle = -\langle \Pi_\mathcal{E} \nabla R(X), W \rangle + O(\|Y\|^2), \quad W \in \mathrm{T}\mathcal{E}.$$

To determine the dynamics for $Y$ is easier: Here, we only need to note that $\langle \Pi_\mathcal{E} \nabla R(X), V \rangle = 0$ for $V \in \mathrm{T}\mathcal{E}^\perp$ arbitrary to arrive at

$$\langle \dot{Y}, V \rangle = -\langle \Pi_\mathcal{L}(R^{\mathrm{aug}})''(X)Y, V \rangle + \mathcal{O}(\|Y\|^2), \quad V \in \mathrm{T}\mathcal{E}^\perp.$$

Now it is only left to note that in the case of partially unitary $E$, $\dot{X} = -\Pi_\mathcal{E} \nabla R(X)$ are the EQUI dynamics. $\qquad\square$

## 4 Experiments

We perform an experiment in order to showcase the difference between a network whose layers obey the compatibility assumption and one where they do not. Additionally, we test training the networks using stochastic gradient descent (SGD) with varying batch sizes, to investigate to which extent our theoretical results (which are only about gradient flow) also apply to them.

The code for the experiment was written in Python, using the PyTorch library. The code can be accessed at github.com/usinedepain/eq_aug_dyn/.

### 4.1 Experiment description

We consider two nominal architectures consisting of convolutional layers with a fully connected last layer, and equivariant pooling and layer normalization layers – see Figure 3. The two architectures differ in the choice of support for the convolutional filters – one is the cross support (denoted by Cross in Figure 4) and one is the skew support (denoted by Skew in Figure 4), as in Figure 2. These were trained in EQUI mode for 250 epochs, manifestly invariant to the rotation action of $\mathbb{Z}_4$, to classify MNIST (LeCun et al., 1998). We used SGD as the optimizer, with an MSE loss with the labels as one-hot vectors; the learning rate was set to $5 \cdot 10^{-4}$ and the batch size was set to 100. For the cross shaped support, we ran two experiments, with partially unitary and non-unitary embedding operator $L$ (denoted by Non-Unitary in Figure 4), in AUG mode, resulting in a total of three experiments. Basis vectors for the non-unitary embedding $L$ were drawn as cross-shaped filters with i.i.d. Gaussian entries. We trained 30 networks in EQUI mode for each configuration. Starting from these in total 90 networks, we then switched mode to AUG for ten more epochs of training with batch sizes 1, 25, and 100, and for 50 more epochs of training with gradient descent, with a learning rate of $2.5 \cdot 10^{-4}$. The reason we lowered the learning rate for this part of the experiment was to make the SGD 'closer' to a flow. The reason we trained for 50 epochs with gradient descent compared with the 10 epochs of training with minibatch SGD is that gradient descent only updates once per pass over the entire data set. The images were normalized with respect to mean and standard deviation before being sent to the first layer.

The purpose of the experiment was to observe how quickly the layers of the different CNNs would drift away from $\mathcal{E}$ when using augmentation. After the EQUI mode epochs, we will likely be close to a stationary point of the equivariant flow. This point will also be stationary for the augmented flow if the compatibility condition holds. By Example 3.6, this is the case for the cross support, but not for the skew support. Hence, we can expect the latter to drift away faster. For the two cross support architectures, we also expect to see a difference, since $\mathcal{E}$ only is an invariant set when using partially unitary $L$, by Theorem 3.7 (part 2). Hence, the non-unitary version should be more prone to drift. Furthermore, since our theory holds for gradient descent, but not necessarily for SGD which can 'jump away' from $\mathcal{E}$, we also expect that we see the least amount of drift when training with gradient descent, and the most drift when training with SGD with a batch size of 1.

### 4.2 Results and analysis

After each batch in the augmented training, the distance of the layers from $\mathcal{E}$ was recorded. In Figure 4, we plot, for each batch size, on the left, the mean distance over each full epoch, and on the right, the distance after each batch in the first epoch. In the top left sub-figure we plot the distance after each epoch of training with gradient descent. In essence, our above hypotheses are confirmed: The architecture with cross support and partially unitary $L$ stays close to $\mathcal{E}$, whereas the others drift. The skew-support architecture drifts more in absolute numbers, but not too much should be read into this – it may very well look different for other nominal architectures. Furthermore, we see that the layers of the networks trained with (non-stochastic) gradient descent stay comparatively close to $\mathcal{E}$. In numbers, the distance for the cross supports after 50 batches of training with a batch size of 100 is on the order of 5e-3, which is quite close to the corresponding

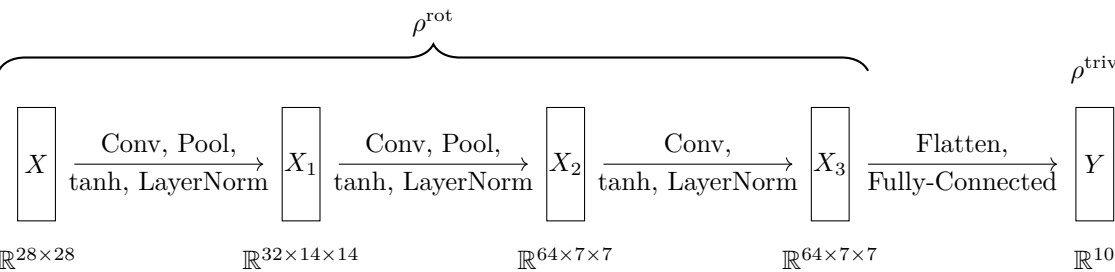

Figure 3: The architecture consists of three convolutional layers with filters $\varphi$ having support as in Figure 2 (left or right), followed by a flattening and then a fully-connected layer.

values for the skew supports as well as for non-unitary embedding, whereas after 50 epochs of training with gradient descent the distance for the cross supports is on the order of 1e-7, which is much smaller than the corresponding values for the skew supports as well as for non-unitary embedding. Also note that the non-unitary cross architectures have some outliers – we believe that the cause for this is numerical problems when initializing the AUG training - this entails solving a linear set of equations $Lc = A_0$, with a potentially ill-conditioned $L$.

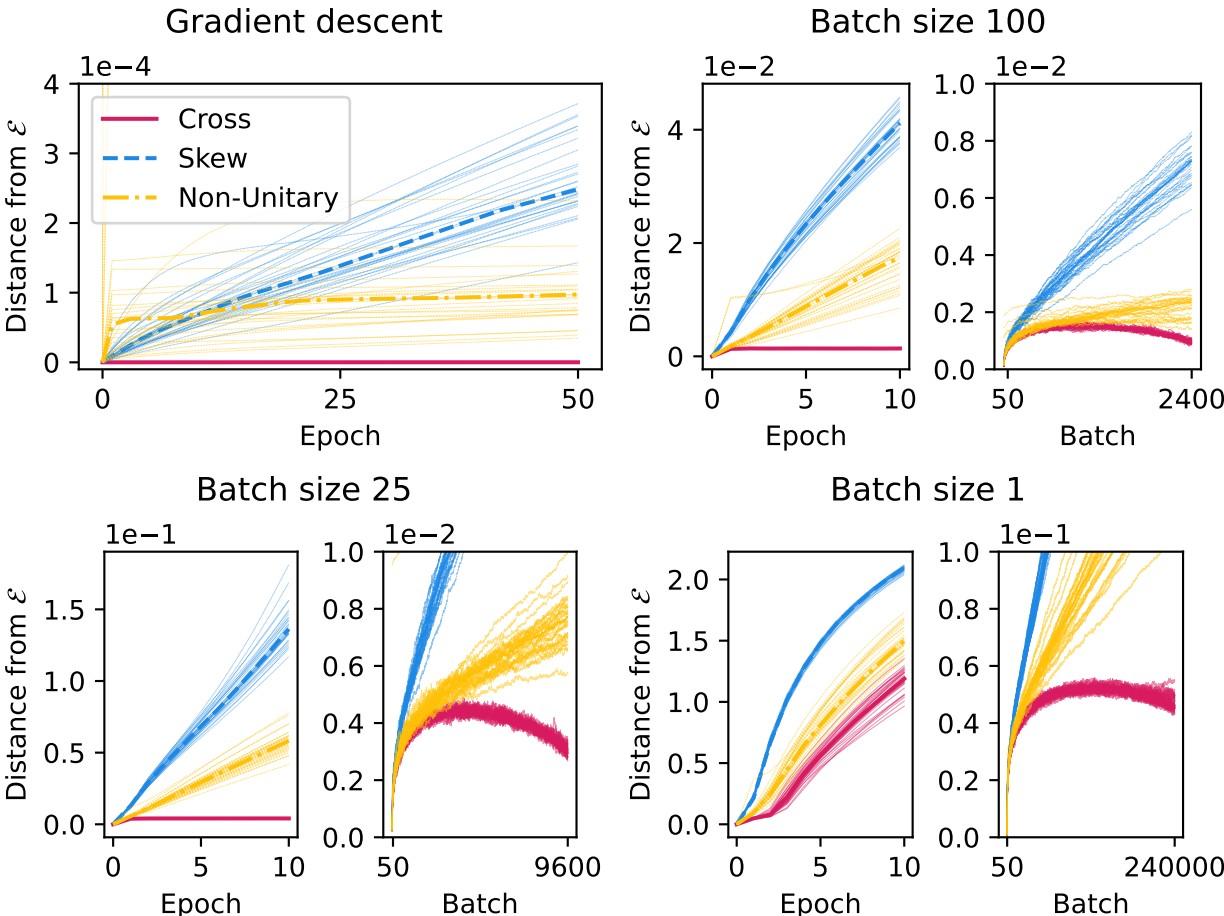

Figure 4: Top left sub-figure (gradient descent): distance in 2-norm of layers from $\mathcal{E}$ per epoch of augmented training.
On the left of the other sub-figures: distance in 2-norm of layers from $\mathcal{E}$ averaged over the batches per epoch of augmented training. On the right of the other sub-figures: distance in 2-norm of layers from $\mathcal{E}$ during the first epoch of augmented training.
The fainter lines are the individual experiments and the thicker lines are the medians over all experiments. The noticeable outliers in faint yellow are most likely due to numerical errors in calculating the non-unitary embedding operator by solving a linear system. Best viewed in color.

The cross shaped architecture in this case seems to be relatively stable for minibatch SGD with batch sizes 25 and 100. It should however be noted that although it appears so in the left figure of the top right and bottom left sub-figures, it does not converge to 0, but rather to around 1e-3. This is not a contradiction – Theorem 3.12 indicates that $\mathcal{E}$ may be unstable for AUG also in the cross-shaped case. The general trend is however as we expected. The experiments indicate that as long as the batch sizes are large enough, the dynamics of stochastic gradient descent are close enough to the gradient descent for our theoretical results to still give a good picture. It is important to note that for SGD with batch size of 1, the cross support also drifts from

$\mathcal{E}$ quite dramatically, differing from the other cases. To investigate for which 'degrees of stochasticity' our theoretical results are still relevant is interesting future work.

In the appendix, we perform another experiment testing another architecture with different symmetry group. In essence, the results are again in accordance with the theory. They reveal connections between the symmetry group, representations and stability of $\mathcal{E}$ that is subject to further theoretical work.

## 5    Conclusion

In this paper we set out to investigate the relationship between the gradient flows of networks equivariant by design and nominal networks trained on augmented data during training. It turns out that the geometry of the spaces of admissible and equivariant layers, $\mathcal{L}$ and $\mathcal{H}_G$ respectively, is key. Under the *compatibility condition* $\Pi_\mathcal{L}\Pi_G = \Pi_G\Pi_\mathcal{L}$, we showed that the stationary points in the equivariant space $\mathcal{E}$ are the same for the equivariant and augmented flows, but that they do not necessarily share the set of equivariant stable points. Furthermore, we showed that $\mathcal{E}$ is invariant for the augmented flow given unitary parametrizations of our layers. In fact, to first order approximation the dynamics of the augmented flow decouples in $\mathcal{E}$ and $\mathrm{T}\mathcal{E}^\perp$ in this case.

### 5.1    Practical take-aways

What practical recommendations can be extracted from our results? One lesson is that in order to promote equivariance through augmentation, one should consider the option of initializing all parameters in an equivariant way. Also, one should check that the compatibility condition holds - as we have seen, this will for most popular, reasonable architectures, already be the case.

Does our work give a definitive answer to whether one should augment the data or restrict the architecture? The short answer is no. Our results can be used to argue both for augmentation and for restriction: First, advocates of either strategy can argue that 'their' strategy have the exact same set of equivariant stationary point, and hence does not 'miss' anything the other strategy can find. 'Restrictioners' can point to their strategy making more points on $\mathcal{E}$ stable, and hence a stronger bias. The latter can however, by 'augmenters', also be argued to be indicative of the restriction strategy inducing more 'bad' local minima on $\mathcal{E}$. These can be escaped, and a better minimum on $\mathcal{E}$ can later be found, via an 'expedition' in $\mathcal{H}\backslash\mathcal{E}$. That expedition is however not guaranteed to find its way back to $\mathcal{E}$. However, our results provide a better understanding of the dynamics occurring in the two strategies, which in of itself is important.

### 5.2    Future work

This work is an initial foray into the effects of data augmentation on the dynamics of gradient flows during training for symmetric tasks. Future work includes further investigating the role of the symmetry group $G$, and extending the results to a stochastic setting.

### Acknowledgement

The authors would like to thank the anonymous reviewers of all versions of this manuscript, whose insightful comments have helped to significantly improve it. This work was partially supported by the Wallenberg AI, Autonomous Systems and Software Program (WASP) funded by the Knut and Alice Wallenberg Foundation. The computations were enabled by resources provided by the National Academic Infrastructure for Supercomputing in Sweden (NAISS) at Chalmers University of Technology partially funded by the Swedish Research Council through grant agreement no. 2022-0672.

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

## A    Incorporating common architectures into the framework

Before describing how some common architectures can be incorporated into the framework, let us point out a general construction. Instead of specifying $\mathcal{L}$ as a whole, we may choose affine spaces $\mathrm{L}(X_i, X_{i+1})$ of operators within each space $\mathrm{Hom}(X_i, X_{i+1})$, and then in the end set $\mathcal{L} := \bigoplus_{i \in [L]} \mathrm{L}(X_i, X_{i+1})$. This in particular makes it clear that many of the constructions below can easily be combined with each other. Note that in the case that an architecture is built by such a direct sum, the orthogonal projector $\Pi_{\mathcal{L}}$ is also applied component-wise, through projecting each layer to the respective tangent space $\mathrm{T}\,\mathrm{L}(X_i, X_{i+1})$. In the layer-wise constructions below we will refer to the vector spaces spaces as $U$ and $V$ instead of $X_i$ and $X_{i+1}$ to simplify notation.

**Fully connected without bias**    Letting $\mathcal{L} = \mathcal{H}$ corresponds to a fully connected MLP layer with no bias.

**Fully connected with bias**    Allowing for bias terms corresponds to using affine instead of linear maps in each layer. Affine maps can however be considered as linear maps on a lifted space: Given spaces $U$ and V, we set $\widetilde{U} = U \oplus \mathbb{R}$, $\widetilde{V} = V \oplus \mathbb{R}$ and define

$$\mathrm{L}(\widetilde{U}, \widetilde{V}) := \left\{ \begin{bmatrix} A & b \\ 0 & 1 \end{bmatrix} \ \middle| \ A \in \mathrm{Hom}(U, V), b \in V \right\}.$$

Extending the nonlinearity $\sigma$ to $\widetilde{\sigma}(x, 1) := (\sigma(x), 1)$, we obtain

$$\widetilde{\sigma}\left(\begin{bmatrix} A & b \\ 0 & 1 \end{bmatrix}\begin{bmatrix} x \\ 1 \end{bmatrix}\right) = \begin{bmatrix} \sigma(Ax + b) \\ 1 \end{bmatrix},$$

which emulates a fully connected layer with bias. Identifying the original data-points $x_i \in U$ with $(x_i, 1) \in U \oplus \mathbb{R}$, we can thus include layers with bias terms into our framework.

Note that representations $\rho_U$, $\rho_V$ also need to be extended: $\widetilde{\rho}_U(g)(x, 1) = (\rho_U(g)x, 1)$, and so on. It is not hard to show that these nonlinearities are equivariant if and only if the original ones are. The equivariance of the affine maps $x \mapsto Ax + b$ are furthermore equivalent to the equivariance of the corresponding linear maps in $\mathrm{Hom}(\widetilde{U}, \widetilde{V})$

**Convolutional layers**  Since the space of convolutional operators $x \mapsto \varphi * x$ is a linear one, we may choose $\mathrm{L}(X_i, X_{i+1})$ equal to it. We can even restrict the filters $\varphi$ to lie an affine space, such as a space of filter having a certain support. Note that every operator in this space already is equivariant with respect to translations, but that equivariance with respect to other groups could still be relevant.

**Residual connections**  Residual layers are layers of the form $U \to U$, $x' = x + \sigma(Ax)$. They can be modelled via introducing an additional intermediate vector space $W = U \oplus U$, and setting

$$\mathrm{L}(U, W) = \left\{ \begin{bmatrix} \mathrm{id} \\ A \end{bmatrix} \,\middle|\, A \in \mathrm{Hom}(U, U) \right\}, \; \widetilde{\sigma}_W(x, y) := (x, \sigma(y)),$$

$$\mathrm{L}(W, U) = \left\{ \begin{bmatrix} \mathrm{id} & \mathrm{id} \end{bmatrix} \right\}, \widetilde{\sigma}_U = \mathrm{id}$$

Now, for $\widetilde{A} \in \mathrm{Hom}(U, W)$ and $E \in \mathrm{Hom}(W, U)$, we have

$$\widetilde{\sigma}_U(E\widetilde{\sigma}_W(\widetilde{A}x)) = \widetilde{\sigma}_U(E\widetilde{\sigma}_W(x, Ax)) = \widetilde{\sigma}_U(E(x, \sigma(Ax))) = \widetilde{\sigma}_U(x + \sigma(Ax)) = x + \sigma(Ax).$$

It should be clear that more general residual connections can be incorporated in a similar manner.

As for group actions, it is in this context natural to assume that the representations of $G$ on the in- and output spaces – only then will id be equivariant. We can then define $\rho_W(g)(x, y) = (\rho(g)x, \rho(g)y)$ – a layer $(\mathrm{id}, A) \in \mathrm{L}(U, W)$ is then equivariant if and only if $A \in \mathrm{L}(U, U)$ is.

**Attention layers**  Attention layers (Vaswani et al., 2017) map sequences $u \in U^n$ to sequences $v \in V^n$, where $U$ and $V$ are vector spaces, through the formula

$$w_i = \sum_j \alpha_{ij}\nu_j, \quad \alpha_{i,j} = \mathrm{softmax}(\langle q_i, k.\rangle)_j,$$

where $\langle q_i, k.\rangle = (\langle q_i, k_\ell \rangle)_{\ell \in [n]}$. Here, $q \in W^n$ and $k \in W^n$ are the so called *query* and *key* sequences, with values in a third vector space $W$, $\nu \in V^n$ is the *value* sequence, and softmax is the function:

$$\mathrm{softmax}(p)_i = \frac{e^{p_i}}{\sum_j e^{p_j}}.$$

Each sequence $(\alpha_{i,.})_{j \in [n]}$ is hence a probability distribution with respect to which one takes an expected value of the values $(\nu_j)_{j \in [n]}$ to produce a new value $w_i$. The queries, keys and values are calculated via applying linear maps $Q : U \to W$, $K : U \to W$ and $N : U \to V$ elementwise, i.e. $q_i = Qu_i$, $k_j = Ku_j$ and $\nu_i = Nu_i$ (there are variations where these maps are non-linear; in this treatment we will assume they are linear). To simplify notation, we write $q = Qu$, $k = Ku$ and $\nu = Nu$.

We can include such attention layers in our framework as follows: First, let us, similarly to above, introduce an intermediate space $Z = W^n \bigoplus W^n \bigoplus V^n$, and set

$$\mathrm{L}(U^N, Z) = \{u \to (Qu, Ku, Nu) \,|\, Q, K \in \mathrm{Hom}(U, W), N \in \mathrm{Hom}(U, V)\},$$

$$\sigma_Z^{\mathrm{att}}(q, k, \nu) = \left(0, 0, \Big(\sum_j \alpha_{ij}\nu_j\Big)_{i \in [n]}\right), \text{ with } \alpha_{i,j} = \mathrm{softmax}(\langle q_i, k.\rangle)_j$$

$$\mathrm{L}(Z, V^n) = \{(q, k, \nu) \to \nu\}, \quad \sigma_{V^n}^{\mathrm{att}} = \mathrm{id}.$$

Note that these definitions may seem unnecessarily complicated, but they make sure that domain and ranges of all non-linearities are the same, which our framework formally requires. It is not hard to check that, in sequence, applying a linear map in $L(U^n, Z)$, $\sigma_Z^{\text{att}}$, the linear map in $L(Z, V^n)$ and finally $\sigma_{V^n}^{\text{att}}$ to a sequence in $U^n$ has the effect of an attentional layer.

As for the group actions, we imagine unitary representations $\varrho_{\text{in}}$ and $\varrho_{\text{out}}$ are given on the input and output-spaces $U$ and $V$, respectively. These are naturally extended to actions on $U^n$ and $V^n$ through $(\rho_U(g)u)_i = \varrho_{\text{in}}(g)u_i$ and $(\rho_V(g)v)_i = \varrho_{\text{out}}(g)v_i$. In order to incorporate the intermediate spaces, we assume that a unitary representation $\varrho_{\text{key}}$ on the key-query space $W$ is given, which in the same vain can be extended to a representation $\rho_W$ on $W^n$. If we then define $\rho_Z(g)(q, k, \nu) = (\rho_W(g)q, \rho_W(g)k, \rho_V(g)\nu)$ on $Z$, $\sigma_Z^{\text{att}}$ becomes equivariant:

$$\sigma_Z^{\text{att}}(\rho_Z(g)(q, k, \nu)) = \sigma_Z^{\text{att}}(\rho_W(g)q, \rho_W(g)k, \rho_V(g)\nu)$$
$$= \left(0, 0, \sum_j \text{softmax}(\langle \varrho_{\text{key}}(g)q_i, \varrho_{\text{key}}(g)k_.\rangle)_j \varrho_{\text{out}}(g)\nu_j\right)$$
$$= \left(0, 0, \varrho_{\text{out}}(g) \sum_j \text{softmax}(\langle q_i, k_.\rangle)_j \nu_j\right)$$
$$= \rho_Z(g)\sigma_Z^{\text{att}}(q, k, \nu),$$

where the unitarity of $\varrho_{\text{key}}$ was used in step three. It should be noted that a map in $\mathrm{L}(U^n, Z)$ is equivariant if and only if $Q$, $K$ and $N$ are. The equivariance of the map in $L(Z, V^n)$ and of $\sigma_{V^n}^{\text{att}}$ is also clear.

SE(3) transformers Let us argue that modulo a few minor technicalities, the prominent SE(3)-transformer (Fuchs et al., 2020) emerges as $G$-equivariant transformers in this way: In this setting, the in and out features are vector fields $f_i : \mathbb{R}^3 \to \mathcal{V}$, where $\mathcal{V}$ is a space on which the special euclidean group SE(3) is acting, of the form $v_i \delta_{x_i}$, where $\delta_{x_i}$ are Dirac deltas for some fixed positions $x_i$. Since SE(3) is not compact, we must restrict the $Q$, $K$ and $N$-maps in $\mathrm{L}(U^n, Z)$ to a priori be convolutions, i.e. equivariant to translations – the symmetry group then becomes SO(3). By further letting the softmax operation occur over neighborhoods of each $i$ in an a-priori given graph, letting $Q$, $K$ and $N$ sample more general vector fields in the positions $x_i$, and adding a self-attention mechanism, i.e. in essence an additional linear residual connection of each feature $f_i$ to itself, we obtain the architecture. Since this section is already unnecessarily technical as it is, we choose to omit the technical details.

*Remark:* It seems that this formulation of an equivariant transformer architecture is novel – the authors have not seen it in this generality in the literature (although, as we have seen, special cases of it have been – we would also like to mention Vector Neurons (Deng et al., 2021)). It is unclear whether this more general formulation can be used to obtain new practically relevant architectures. We deem the detailed investigation of this matter beyond the scope of this work.

**Message passing**   A popular way to process graph data are so-called *message passing networks*. Let's discuss a simple version of them here – linear message maps. In one sense, they are simplified attention networks: Again, features $u_i \in U$ on nodes of a graph are processed by the same linear maps $S : U \to V$ and $I : U \to V$, and then mixed:

$$v_i = Su_i + \sum_j a_{ij} I u_j.$$

The only difference is that the coupling coefficients $a_{ij}$ here are constant, given by a weighted adjacency matrix of the graph. Note that we here include a separate 'self-connection map' $S$ – this would correspond to self-attention above. Note that message passing layers are inherently equivariant towards permutations (of both the features and the adjacency matrix, corresponding to a reordering of the graph labels).

Because of their similarities to attention layers, we can realize them in a similar manner as above. The input space consists of pairs of feature sequences and adjacency matrix, say $\mathcal{U} = U^n \bigoplus \mathbb{R}^{n,n}$, and the output space

is similarly $\mathcal{V} = V^n \bigoplus \mathbb{R}^{n,n}$. We define an intermediate space $\mathcal{Z} = V^n \bigoplus V^n \bigoplus \mathbb{R}^{n,n}$ and define

$$\mathrm{L}(\mathcal{U}, \mathcal{Z}) = \{(u, A) \mapsto (Su, Iu, A) \,|\, S, I \in \mathrm{Hom}(U, V)\}, \quad \sigma_{\mathcal{Z}}^{\mathrm{MP}}(s, \iota, A) = \Big(0, (s_i + \sum_j A_{ij}\iota_j)_i, A\Big)$$

$$\mathrm{L}(\mathcal{Z}, \mathcal{V}) = \{(s, \iota, A) \mapsto (\iota, A)\}, \quad \sigma_{\mathcal{V}}^{\mathrm{MP}} = \mathrm{id}\,.$$

By slightly changing the formulations, other versions of message passing could also easily be treated, i.e. versions with non-linear message maps or versions with edge-features.

Representations acting separately on each feature $u_i$, as in the transformer example, can immediately be included into this framework. Let us also note that regular representations with respect to subgroups of the permutation group $S_n$ naturally can be included. To be more precise, let $G \subseteq S_n$ be a subgroup, and $\varrho_{\mathrm{in}}$, $\varrho_{\mathrm{out}}$ be representations of $G$ on $U$ and $V$, respectively. We can then accomodate actions on $U^n$, $V^n$ and $\mathbb{R}^{n,n}$ of the following form:

$$(\rho_U(\pi)u)_i = \varrho_{\mathrm{in}}(\pi)u_{\pi^{-1}(i)}, \quad (\rho_V(\pi)v)_i = \varrho_{\mathrm{out}}(\pi)v_{\pi^{-1}(i)}, \quad (\rho_{\mathrm{adj}}(\pi)A)_{i,j} = A_{\pi^{-1}(i)\pi^{-1}(j)}$$

These, of course, induce natural representations $\rho_{\mathcal{U}}$, $\rho_{\mathcal{V}}$ of $G$ on $\mathcal{U}$ and $\mathcal{V}$. If we define the representation on $\mathcal{Z}$ as $\rho_{\mathcal{Z}}(\pi)(s, \iota, A) = (\rho_{\mathcal{U}}(\pi)s, \rho_{\mathcal{V}}(\pi)\iota, \rho_{\mathrm{adj}}(\pi)A)$, $\sigma_{\mathcal{Z}}^{\mathrm{MP}}$ becomes equivariant. We have

$$\sigma_{\mathcal{Z}}^{\mathrm{MP}}(\rho_{\mathcal{Z}}(\pi)(s, \iota, A)) = \sigma_{\mathcal{Z}}^{\mathrm{MP}}(\rho_{\mathcal{V}}(\pi)s, \rho_{\mathcal{V}}(\pi)\iota, \rho_{\mathrm{adj}}(\pi)A) = \Big(0, \big((\rho_{\mathcal{V}}(\pi)s)_i + \sum_j (\rho_{\mathrm{adj}}(\pi)A)_{ij}(\rho_{\mathcal{V}}(\pi)\iota)_j\big)_i, \rho_{\mathrm{adj}}(\pi)A\Big)$$

We calculate

$$(\rho_{\mathcal{V}}(\pi)s)_i + \sum_j (\rho_{\mathrm{adj}}(\pi)A)_{ij}(\rho_{\mathcal{V}}(\pi)\iota)_j = \varrho_{\mathrm{out}}(\pi)s_{\pi^{-1}(i)} + \sum_j A_{\pi^{-1}(i)\pi^{-1}(j)}\iota_{\pi^{-1}(j)}$$

$$= \varrho_{\mathrm{out}}(\pi)\big(s_{\pi^{-1}(i)} + \sum_j A_{\pi^{-1}(i)j}\iota_j\big) = \rho_{\mathcal{V}}(\pi)\big((s_i + \sum_j A_{ij}\iota_j)_i\big)_i\,.$$

Hence,

$$\sigma_Z^{\mathrm{MP}}(\rho_{\mathcal{Z}}(\pi)(s, \iota, A)) = \Big(\rho_{\mathcal{V}}(\pi)0, \rho_{\mathcal{V}}(\pi)\big((s_i + \sum_j A_{ij}\iota_j)\big), \rho_{\mathrm{adj}}(\pi)A\Big) = \rho_{\mathcal{Z}}(\pi)\sigma_Z^{\mathrm{MP}}(s, \iota, A).$$

Also, the only operator in $\mathrm{L}(\mathcal{Z}, \mathcal{V})$ and $\sigma_{\mathcal{V}}^{\mathrm{MP}}$ trivially become equivariant.

**Recurrent architectures**  In recurrent architectures, the same layers are applied several times. Since constraining layers to be equal is a linear operation, they can easily be included in our framework. For instance, the recurrent architecture

$$x_{i+1} = \sigma_i(Mx_i), \quad i \in [L]$$

where all $A_i$ equal a common linear map $M$, can be described through

$$\mathcal{L} = \{A \in \mathcal{H} \,|\, A_i = A_{i+1}, i \in [L]\}.$$

# B The compatibility condition

Here, we discuss the compatibility condition a bit more at length than in the main paper. In many of the examples presented in the previous section, the compatibility condition is immediate due to Proposition 3.4:

- Fully connected with bias.

- Residual connections.

- Attention layers.

- Message passing layers.

- Recurrent architectures (when all $\rho_i$ are equal).

In all of these cases, for the types of representations discussed above, it is not hard to show that all operators $\overline{\rho}(g)$ leaves the space $\mathcal{L}$ invariant. Generally and intuitively speaking, this is due to the 'linear parts' of the maps being unrestricted. As an example, let us carry out the details for the residual connections. For a map $\Lambda = [\mathrm{id}, M]^T$, we have

$$\overline{\rho}(g)\Lambda = [\rho(g)\,\mathrm{id}\,\rho(g)^{-1}, \rho(g)A\rho(g)^{-1}] = [\mathrm{id}, \rho(g)A\rho(g)^{-1}],$$

so that $\overline{\rho}(g)\Lambda$ is again a map in $\mathrm{Hom}(U, V)$, since $\rho(g)A\rho(g)^{-1}$ still is linear. The other cases are dealt with similarly.

Now let us describe a few more slightly more involved examples, beginning with the case of convolutions supported on non-symmetric $\Omega$.

**Convolutions with skew support**   Let us finish the discussion on $\mathcal{L}_\Omega$ begun in Example 3.6, beginning with a description of the orthogonal projection onto $\mathcal{L}_\Omega$.

**Lemma B.1.** *Let $P_\Omega$ be the linear map that maps convolutional operators with filter $\varphi$ to the convolutional operator with filter $\mathbb{1}_\Omega \cdot \varphi$, i.e. zeroing out indices outside $\Omega$, and is zero on $\mathcal{C}^\perp$. Then,*

$$\Pi_{\mathcal{L}_\Omega} = P_\Omega \Pi_{\mathcal{C}}.$$

*Proof.* Let us first show that $P_\Omega \Pi_{\mathcal{C}}$ defines a projection, i.e. that $P_\Omega \Pi_{\mathcal{C}} P_\Omega \Pi_{\mathcal{L}} = P_\Omega \Pi_{\mathcal{C}}$. This is not hard – applying $\Pi_{\mathcal{C}}$ to an $A$ yields a convolutional operator, say $C_\varphi$, which is then mapped to $C_{\mathbb{1}_\Omega \cdot \varphi}$ by $P_\Omega$. This operator is however not changed by $\Pi_C$ (since it is convolutional), and multiplying the filter with $\mathbb{1}_\Omega$ again does not change it. Hence, $P_\Omega \Pi_{\mathcal{C}} P_\Omega \Pi_{\mathcal{L}} A = C_{\mathbb{1}_\Omega \cdot \varphi} = P_\Omega \Pi_{\mathcal{C}} A$.

Now let us show that it is orthogonal. For this, we need to show that $\langle A - P_\Omega \Pi_{\mathcal{C}} A, P_\Omega \Pi_{\mathcal{C}} B \rangle = 0$ for all $A$ and $B$. Let us first notice that $\langle A - \Pi_{\mathcal{C}} A, P_\Omega \Pi_{\mathcal{C}} B \rangle = 0$, since $A - \Pi_{\mathcal{C}} A \in \mathcal{C}^\perp$ and $P_\Omega \Pi_{\mathcal{C}} B \in \mathcal{C}$. It hence suffices to show that $\langle \Pi_{\mathcal{C}} A - P_\Omega \Pi_{\mathcal{C}} A, P_\Omega \Pi_{\mathcal{C}} B \rangle = 0$. Let us first calculate $\langle C_\varphi, C_\psi \rangle$ for two given filters. We have by definition

$$\langle C_\varphi, C_\psi \rangle = \sum_{i \in [N]^2} \langle C_\varphi e_i, C_\psi e_i \rangle,$$

where $e_i$ is the canonical basis of $\mathbb{R}^{N,N}$. We have

$$(C_\varphi e_i)[\ell] = \sum_{k \in [N]^2} e_i[\ell - k]\varphi(k) = \varphi[\ell - i],$$

since $e_i[\ell - k] = 1$ precisely when $k = \ell - i$ (and zero otherwise). Hence

$$\sum_{i \in [N]^2} \langle C_\varphi e_i, C_\psi e_i \rangle = \sum_{i \in [N]^2} \varphi[\ell - i]\psi[\ell - i] = N^2 \langle \varphi, \psi \rangle.$$

Now, writing $\Pi_{\mathcal{C}} A = C_\varphi$ and $\Pi_{\mathcal{C}} B = C_\psi$, we get

$$\langle C_\varphi - P_\Omega C_\varphi, P_\Omega C_\psi \rangle = \langle C_{\mathbb{1}_{\Omega^c} \cdot \varphi}, C_{\mathbb{1}_\Omega \cdot \psi} \rangle = N^2 \langle \mathbb{1}_{\Omega^c} \cdot \varphi, \mathbb{1}_\Omega \cdot \psi \rangle = 0,$$

since the filters in the last expression have disjoint supports.

It remains to show that the range of $P_\Omega \Pi_{\mathcal{C}}$ is equal to $\mathcal{L}_\Omega$. But this is easy – it is clear that the operator maps any operator to a filter operator with filter supported on $\Omega$, i.e. into $\Omega$, and that it does not do anything to an operator already there. □

Let us now show that $\Pi_{Z_4}$ acts directly on the filter for a convolutional operator.

**Lemma B.2.** *For $C_\varphi \in \mathcal{C}$ and the rotation action of $\mathbb{Z}_4$, we have $\Pi_{\mathbb{Z}_4} C_\varphi = C_{\Pi_{\mathbb{Z}_4}\varphi}$.*

*Proof.* This follows from the twirling formula (Lemma 3.5) and the fact that the representations act on the filter. We have

$$\Pi_{\mathbb{Z}_4} C_\varphi = \int_{\mathbb{Z}_4} \bar{\rho}(g) C_\varphi \, \mathrm{d}\mu(g) = \int_{\mathbb{Z}_4} C_{\rho(g)\varphi} \, \mathrm{d}\mu(g) = C_{\int_{\mathbb{Z}_4} \rho(g)\varphi \, \mathrm{d}\mu(g)} = C_{\Pi_{\mathbb{Z}_4}\varphi},$$

where the penultimate step follows by linearity (note that here the integral is actually a sum). $\qquad\square$

Now we can construct an operator for which the compatibility condition $\Pi_{\mathcal{L}_\Omega}\Pi_{\mathbb{Z}_4} = \Pi_{\mathbb{Z}_4}\Pi_{\mathcal{L}_\Omega}$ is not fulfilled. Consider a convolutional operator $C_\varphi$ given by a filter with support as in Figure 2 (right) where the top left element is non-zero. For such an operator we have that applying $\Pi_{\mathcal{L}_\Omega}$ to it does not change it at all. Subsequently applying $\Pi_{\mathbb{Z}_4}$ is by Lemma B.2 the same as averaging rotations of the filters. Since we have assumed that the top left element is non-zero, this yields a filter where all corner elements are non-zero. An operator given by such a filter is surely not in $\mathcal{L}_\Omega$, which $\Pi_{\mathcal{L}_\Omega}\Pi_{\mathbb{Z}_4} C_\varphi$ is.

**Diagonal operators**   Let us discuss a case in which Proposition 3.4 does not apply, but the compatibility condition is still satisfied. We consider a single layer, let the input and output space be equal to $\mathbb{R}^3$, and $G = \mathrm{O}(3)$ be the orthogonal group acting in the canonical way on both spaces. It is then not hard to realize that $\mathcal{H}_G = \mathrm{span\ id}$ (shooting flies with cannons, one can appeal to $\mathbb{R}^3$ being an irrep of the action, and then to Schur's Lemma, but one can also deduce this by more elementary means).

Now consider the architecture choice of only using diagonal operators as linear layers:

$$\mathcal{L}_{\mathrm{diag}} = \{A \in \mathrm{Hom}(\mathbb{R}^3, \mathbb{R}^3) \,|\, A_{ij} = 0, i \neq j\}.$$

This corresponds to treating the 'channels' of the in-vectors $x$ separately. Then, $\mathcal{L}$ is not invariant under all $\bar{\rho}(g)$ – the eigenvectors of the map $\bar{\rho}(g)D$ for a $D \in \mathcal{L}_{\mathrm{diag}}$ with non-constant diagonal are given by $\bar{\rho}(g)e_0, \bar{\rho}(g)e_1, \bar{\rho}(g)e_2$, which are surely not always equal to $e_0, e_1, e_2$, which is necessary for $\bar{\rho}(g)D$ to be diagonal. However, the compatibility condition is satisfied: since $\mathcal{H}_G = \mathrm{span\ id} \subseteq \mathrm{T}\mathcal{L}$, $\Pi_{\mathcal{L}}\Pi_G = \Pi_G\Pi_{\mathcal{L}} = \Pi_G$.

**Subgroups**   Consider any group $G$ and choose $H$ as a subgroup of $G$. For vector spaces $U, V$ on which $G$ is acting, now set

$$\mathcal{L} = \mathcal{H}_H = \{A \in \mathcal{H} \,|\, A\rho_U(h) = \rho_V(h)A, h \in H\},$$

i.e. the linear operators that are only equivariant to the action of the subgroup. $\mathcal{L}$ is then in general not invariant to the action of $G$. The diagonal operators above is in fact an example of this – here, $G = \mathrm{O}(3)$, and the subgroup is the group of permutations of components.

Note that just as in the special case of diagonal operators, the compatibility condition is always satisfied, since $\mathcal{L} = \mathcal{H}_H \supseteq \mathcal{H}_G$.

**Upper triangular matrices and shifts**   Consider as input and output space $U = V = \mathbb{R}^N \bigoplus \mathbb{R}^N \sim \mathbb{R}^{2N}$ and the action of $S_2 = \{\mathrm{id}, \varsigma\}$ on this space by interchanging the coordinates, i.e $\rho(\varsigma)(x, y) = (y, x)$. Let $\mathcal{L}$ be the space of linear maps described by upper triangular matrices, i.e.,

$$\mathcal{L} = \{K \in \mathrm{Hom}(\mathbb{R}^{2N}, \mathbb{R}^{2N}) \,|\, K_{ij} = 0, i > j\}.$$

Alternatively, each element in $\mathcal{L}$ can be written as a block matrix

$$\begin{bmatrix} A^\triangledown & B \\ 0 & C^\triangledown \end{bmatrix}, \quad A, B, C \in \mathbb{R}^{N,N}$$

where we introduced the notation $A^\triangledown$ for the matrix obtained by putting all elements in $A$ under the diagonal to zero. This type of layer can be though of as 'causal' – if we interpret $u = (x, y)$ as a time series

$y_{n-1}, y_{n-2}, \ldots y_0, x_{n-1}, \ldots$, the value $A(u)[n-i]$ only depends on the first $i$ values in $u$, i.e., the values 'before' time $i$.

To see what $\mathcal{E}$ is in this setting, let us notice that the lifted representation of $S_2$ on $\mathrm{Hom}(U, V)$ is as follows:

$$\bar{\rho}(\varsigma)\left(\begin{bmatrix} A & B \\ D & C \end{bmatrix}\right) = \rho(\varsigma) \begin{bmatrix} A & B \\ D & C \end{bmatrix} \rho(\varsigma)^{-1} = \begin{bmatrix} 0 & \mathrm{id} \\ \mathrm{id} & 0 \end{bmatrix} \begin{bmatrix} A & B \\ D & C \end{bmatrix} \begin{bmatrix} 0 & \mathrm{id} \\ \mathrm{id} & 0 \end{bmatrix} = \begin{bmatrix} C & D \\ B & A \end{bmatrix} \tag{12}$$

Thus, a map is in $\mathcal{H}_G$ if and only if $A = C$ and $B = D$. If it should also be in $\mathcal{L}$, $A$ and $C$ need to be upper triangular, and $B = D = 0$. Thus

$$\mathcal{E} = \left\{ \begin{bmatrix} A^\triangledown & 0 \\ 0 & A^\triangledown \end{bmatrix} \,\middle|\, A \in \mathbb{R}^{N,N} \right\}$$

We claim that the compatibility condition is not satisfied for this setting. First, $\mathcal{L}$ is clearly not invariant under the action of $S_2$, so that Proposition 3.4 does not apply. By appealing to (12) and the twirling formula, we obtain:

$$\Pi_{S_2} \begin{bmatrix} A & B \\ D & C \end{bmatrix} = \begin{bmatrix} \frac{1}{2}(A+C) & \frac{1}{2}(B+D) \\ \frac{1}{2}(B+D) & \frac{1}{2}(A+C) \end{bmatrix}$$

It is further trivial to see that

$$\Pi_{\mathcal{L}} \begin{bmatrix} A & B \\ D & C \end{bmatrix} = \begin{bmatrix} A^\triangledown & B \\ 0 & C^\triangledown \end{bmatrix}.$$

Hence,

$$\Pi_{\mathcal{L}} \Pi_{S_2} \begin{bmatrix} A & B \\ D & C \end{bmatrix} = \Pi_{\mathcal{L}} \begin{bmatrix} \frac{1}{2}(A+C) & \frac{1}{2}(B+D) \\ \frac{1}{2}(B+D) & \frac{1}{2}(A+C) \end{bmatrix} = \begin{bmatrix} \frac{1}{2}(A+C)^\triangledown & \frac{1}{2}(B+D) \\ 0 & \frac{1}{2}(A+C)^\triangledown \end{bmatrix}, \text{ and}$$

$$\Pi_{S_2} \Pi_{\mathcal{L}} \begin{bmatrix} A & B \\ D & C \end{bmatrix} = \Pi_{S_2} \begin{bmatrix} A^\triangledown & B \\ 0 & C^\triangledown \end{bmatrix} = \begin{bmatrix} \frac{1}{2}(A+C)^\triangledown & \frac{1}{2}B \\ \frac{1}{2}B & \frac{1}{2}(A+C)^\triangledown \end{bmatrix}.$$

These are for $B \neq 0$ not equal, and thus the compatibility condition is not fulfilled.

## C  The converse of Theorem 3.12.1 does not hold

We construct an architecture for which a point $A \in S^{\mathrm{eqv}} = S^{\mathrm{aug}}$ is stable for EQUI, but not for AUG.

- **Architecture** The model is a simple one-layer network:

$$\Psi_\lambda : \mathbb{R}^N \to \mathbb{R}, x \mapsto e^{\langle \lambda, x \rangle},$$

  where $\lambda$ can be chosen freely in $\mathbb{R}^N$. This corresponds to letting $X = \mathbb{R}^N$, $Y = \mathbb{R}$ and $\sigma(z) = e^z$. The vector $\lambda \in \mathbb{R}^N$ can be viewed as $A_0 \in \mathrm{Hom}(\mathbb{R}^N, \mathbb{R})$ and $\mathcal{L} = \mathcal{H}$.

- **Symmetry group** We consider the group of translations $\mathbb{Z}_N$ acting canonically through $\rho^{\mathrm{tr}}$ on $\mathbb{R}^N$ and trivially on $\mathbb{R}$. Then, $\mathcal{E} = \mathcal{H}_{\mathbb{Z}_N} = \mathrm{span}\, \mathbb{1}$.

- **Loss** We use the following loss:

$$\ell(y, y') = -yy'.$$

- **Data** We consider a finite dataset $(x_i, y_i) \in \mathbb{R}^N \times \mathbb{R}$, $i \in [m]$ with the property

$$\frac{1}{m} \sum_{i \in [m]} y_i x_i = 0. \tag{13}$$

  Furthermore, the dataset has the property that all $x_i$ are either in $\mathcal{E}$ or in $\mathcal{E}^\perp$. Say, $x_i$, $i \in I$ are in $\mathcal{E}$ and $x_j$, $j \in J$ are in $\mathcal{E}^\perp$. The corresponding labels satisfy $y_i < 0$, for all $i \in I$ and $y_j > 0$, for all $j \in J$. We also assume that $\mathcal{E} = \mathrm{span}\,\{x_i \,|\, i \in I\}$, and that $J$ is nonempty.

Both $\ell$ and $\sigma$ are equivariant, so that Assumptions 1–2 are all satisfied, as is trivially the compatibility condition. We will now show that $\lambda = 0$ (which is in $\mathcal{E}$) is a stable stationary point for EQUI, but not for AUG. The nominal risk is given by $R(\lambda) = -\frac{1}{m} \sum_{i \in [m]} y_i \sigma(\langle \lambda, x_i \rangle)$. By the chain rule, the gradient for the nominal risk is given by

$$\nabla R(\lambda) = -\frac{1}{m} \sum_{i \in [m]} y_i \sigma'(\langle \lambda, x_i \rangle) x_i.$$

Since $\lambda = 0$, it follows that $\sigma'(\langle \lambda, x_i \rangle) = 1$. Then $\nabla R(\lambda) = 0$, by (13). Thus, $\lambda$ is a stationary point for NOM and thus also for AUG and EQUI, by Theorem 3.7.

Again by the chain rule,

$$R''(\lambda) = -\frac{1}{m} \sum_{i \in [m]} y_i \sigma''(\langle \lambda, x_i \rangle) x_i x_i^*.$$

Now, let us show that for $v \in \mathcal{E}$, $R''(\lambda)[v, v] > 0$. By the proof of Theorem 3.12, this is enough to show that $\lambda = 0$ is a stable point for EQUI. But because of the structure of the dataset we have

$$R''(0)[v, v] = -\frac{1}{m} \sum_{i \in [m]} y_i \sigma''(0) \langle v, x_i \rangle^2 = -\frac{1}{m} \sum_{i \in I} y_i \sigma''(0) \langle v, x_i \rangle^2 > 0,$$

since $y_i < 0$ for $i \in I$. To show that $\lambda = 0$ is unstable for AUG we show that there exists $w \in \mathcal{E}^\perp$ such that $(R^{\text{aug}})''(0)[w, w] < 0$. By Lemma 3.11, we have for $w \in \mathcal{E}^\perp$ that

$$(R^{\text{aug}})''(0)[w, w] = -\int_{\mathbb{Z}_N} \frac{1}{m} \sum_{i \in [m]} y_i \sigma''(0) \langle \rho^{\text{tr}}(k) w, x_i \rangle^2 \, d\mu(k).$$

By unitarity of $\rho^{\text{tr}}$ we have that $\rho^{\text{tr}}(k) w \in \mathcal{E}^\perp$. Thus, the sum above is only over $i \in J$, so that all terms are non-negative. By choosing $w = x_j$ for some $j \in J$, we ensure that at least one term is strictly positive. Thus, $(R^{\text{aug}})''(0)[w, w] < 0$ for that $w$, and $\lambda$ is unstable for AUG.

## D Proofs

Here, we present proofs left out in the main text.

***Proof of Lemma 3.3.*** If $\Pi_\mathcal{L} \Pi_G = \Pi_\mathcal{E}$, $\Pi_\mathcal{L} \Pi_G$ is self-adjoint as an orthogonal projection. Consequently, $\Pi_\mathcal{L} \Pi_G = (\Pi_\mathcal{L} \Pi_G)^* = \Pi_G^* \Pi_\mathcal{L}^* = \Pi_G \Pi_\mathcal{L}$, i.e., $\Pi_G$ and $\Pi_L$ commute.

If we on the other hand assume that $\Pi_G$ and $\Pi_\mathcal{L}$ commute, $\Pi_\mathcal{L} \Pi_G$ becomes self-adjoint. It furthermore is idempotent, since

$$(\Pi_\mathcal{L} \Pi_G)^2 = \Pi_\mathcal{L} \Pi_G \Pi_\mathcal{L} \Pi_G = \Pi_\mathcal{L} \Pi_G^2 \Pi_\mathcal{L} = \Pi_\mathcal{L} \Pi_G \Pi_\mathcal{L} = \Pi_\mathcal{L}^2 \Pi_G = \Pi_\mathcal{L} \Pi_G.$$

Hence, $\Pi_\mathcal{L} \Pi_G$ is an orthogonal projection. Its range is included in both $T\mathcal{L}$ and $\mathcal{H}_G$ – the former is clear, and the latter follows from $\Pi_\mathcal{L} \Pi_G = \Pi_G \Pi_\mathcal{L}$. Hence, the range is included in $T\mathcal{E}$. However, if $A \in T\mathcal{E}$, it must be $\Pi_\mathcal{L} \Pi_G A = \Pi_\mathcal{L} A = A$, since $A \in \mathcal{H}_G$ and $A \in T\mathcal{L}$. Hence, the range of the operator is in fact equal to $T\mathcal{E}$, and we are done. $\qquad\square$

***Proof of Lemma 3.5.*** To simplify the exposition of the argument, let us define the operator $P$ by

$$PA = \int_G \overline{\rho}(g) A \, d\mu(g). \tag{14}$$

It needs to be shown that $P = \Pi_G$. To do so, we first need to show that $PA \in \mathcal{H}_G$ for any $A \in \mathcal{H}$. To do this, it suffices to prove that $\overline{\rho}(g)PA = PA$ for any $g \in G$. Using the fact that $\overline{\rho}$ is an representation of $G$, and the invariance of the Haar measure, we obtain

$$\overline{\rho}(g)PA = \int_G \overline{\rho}(g)\overline{\rho}(h)A \, \mathrm{d}\mu(h) = \int_G \overline{\rho}(gh)A \, \mathrm{d}\mu(h) = \int_G \overline{\rho}(h')A \, \mathrm{d}\mu(h') = PA.$$

Next, we need to show that $PA = A$ for any $A \in \mathcal{H}_G$. But since $\overline{\rho}(g)A = A$ for such $A$, we immediately obtain

$$PA = \int_G \overline{\rho}(g)A \, \mathrm{d}\mu(g) = \int_G A \, \mathrm{d}\mu(g) = A.$$

Consequently, $P : \mathcal{H} \to \mathcal{H}_G$ is a projection. Finally, to establish that $P$ is also orthogonal, we need to show that $\langle A - PA, B \rangle = 0$ for all $A \in \mathcal{H}$, $B \in \mathcal{H}_G$. This is a simple consequence of the unitarity of $\overline{\rho}$ and the fact that $\overline{\rho}(g)B = B$ for all $B \in \mathcal{H}_G$ and $g \in G$:

$$\langle PA, B \rangle = \int_G \langle \overline{\rho}(g)A, B \rangle \, \mathrm{d}\mu(g) = \int_G \langle \overline{\rho}(g)A, \overline{\rho}(g)B \rangle \, \mathrm{d}\mu(g) = \int_G \langle A, B \rangle \, \mathrm{d}\mu(g) = \langle A, B \rangle,$$

which completes the proof that $P = \Pi_G$. $\qquad\square$

## E    Experiments

In this appendix we present additional information about the experiment in the main paper, and also describe an additional experiment we conducted.

### E.1    Experiment details: Compatibility condition and unitarity of embedding operator

**Architecture**    The architecture (Figure 3) used for this experiment consisted of three convolutional layers, followed by a fully-connected linear layer. The structure of the first two convolutional layers was as follows. A convolution with a $3 \times 3$ filter and zero padding with one layer of zeroed border pixels (zero padding destroys any translational equivariance, but not the rotational equivariance), followed by average pooling with a $2 \times 2$ pooling window and a stride of 2, followed by a tanh activation function, lastly followed by a layer normalization. The structure of the third convolutional layer was the same as the first two, without the average pooling. The first convolutional layer had one channel in and 32 channels out, the second had 32 channels in and 64 channels out, and the third had 64 channels in and 64 channels out. In the linear layer we flatten the parameter tensor into a vector in $\mathbb{R}^{3136}$ and map it linearly into $\mathbb{R}^{10}$ with a fully-connected layer.

As discussed earlier any activation function (e.g. tanh) applied pixel-wise to an image will be equivariant to rotations by $\pi/2$ radians. It follows that average pooling with a $2 \times 2$ window and a stride of 2 applied to a square image of even dimensions is also equivariant to rotations. This is true since any even dimensional square pixel grid can be subdivided into some number of $2 \times 2$ grids which will rotate along with the whole grid. Note that rotations of pixels within an individual sub grid does not change the average value of its elements. See also Figure 5.

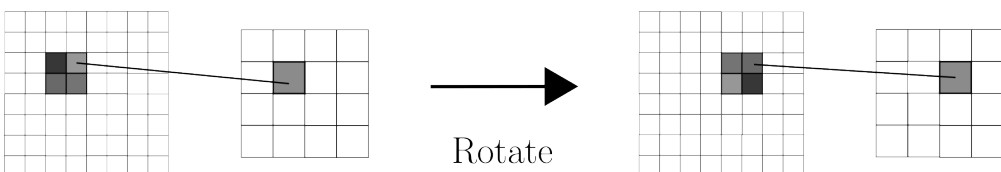

Figure 5: Average pooling with a $2 \times 2$ window and a stride of 2 is equivariant to rotations of square images of even size.

The layer normalization (Ba et al., 2016) technique can be described as follows. Given features $x_i$ $i \in [m]$ in some vector space $U$, the normalized features are

$$\overline{x}_i = x_i - \tfrac{1}{m} \sum_{j \in [m]} x_j.$$

This operation is by linearity equivariant to any representation.

**Hardware**  The $30 \cdot 3 \cdot 4$ trials of the experiment were run in parallel on a super computer using NVIDIA Tesla T4 GPUs with 16GB RAM. We estimate the total computation time to be about 315 hours.

### E.2  An additional experiment

A natural question to ask is if there are differences in the behaviour of the AUG model depending on which symmetry group is used. To this end, we perform an additional experiment. Here, we let the NOM architecture consist of fully connected layers without bias – i.e. $\mathcal{L} = \mathcal{H}$ (making the compatibility condition trivial), taking images as input, say of size $N \times N$. Before being sent to the first layer, the images are normalized by subtracting .5 from every pixel in the image. The number of channels in the early layers are then 1, 32 and 32, respectively. The non-linearities are here chosen as leaky ReLU's, except for the last one which is a SoftMax, and we use a cross-entropy loss. Note that all of these non-linear operations are equivariant to any representation acting pixelwise on images. A detailed sketch of the architecture is given in Figure 6 .

We now consider four different symmetry groups on $\mathbb{R}^{N,N}$

- TRANS  $\mathbb{Z}_N^2$ acting through translations, i.e. through $\rho^{\text{tr}}$.

- ROT $\mathbb{Z}_4$ acting through rotations, i.e. through $\rho^{\text{rot}}$.

- ONEDTRANS $\mathbb{Z}_N$ acting through translations in the $x$-direction, i.e.

$$(\rho^{\text{tr}_0}(k)x)_{i,j} = x_{i-k,j} \tag{15}$$

- TRANSROT The semi-direct product $\mathbb{Z}_N^2 \rtimes \mathbb{Z}_4$ acting through

$$\rho^{\text{trot}}(\iota, k)x = \rho^{\text{rot}}(k)\rho^{\text{tr}}(\iota)x, \quad \iota \in \mathbb{Z}_N^2, k \in \mathbb{Z}_4. \tag{16}$$

  This can, in the same way $\mathbb{Z}_4$ is a discretization of the full group SO(2) of rotations in the plane, be thought of as a discretization of the group SE(2) of isometries in the plane.

These actions induce actions on the early layers $(\mathbb{R}^{N,N})^m$. We let all groups act trivially on the late layers.

We now initialize each architecture on a random point in $\mathcal{E}$, and train each architecture in NOM, AUG and EQUI mode from there – note that we in contrast to the experiment in the main paper do not first train the models in EQUI mode. We only use partially unitary $L$, and here use gradient descent, i.e. accumulate the gradient over the entire epoch before updating the layer parameters. We train the models on MNIST modifieid it in two ways to keep the experiments light. First, we train our models only on the 10000 test examples (instead of the 60000 training samples). Secondly, we subsample the $28 \times 28$ images to images of size $14 \times 14$ (and hence set $N = 14$) using `opencv`'s Bradski (2000) built-in RESIZE function. This simply to reduce the size of the networks. Note that the size of the early (non-equivariant) layers of the model are proportional to the $N^4$, making this downsizing justifiable.

We run each experiment 30 times on Tesla A40 GPUs situated on a cluster, resulting in about 80 hours of GPU time. In Figure 7, we plot the evolution of the values $\|A_{\mathcal{E}\perp}\|$ against the evolution of $\|A - A^0\|$. The opaque lines in each plot is formed by the average values for all thirty runs, whereas the fainter lines are the 30 individual runs. We see that $\mathcal{E}$ is not stable under the augmented flow in any of the experiments, but that the augmented flows stay closer to $\mathcal{E}$ than the nominal ones – this is consistent with our theoretical results.

Note that the scales in the figures are different, to assess the amount the augmented model drifts *relative* to the other ones. We have chosen the limits for the axes as follows:

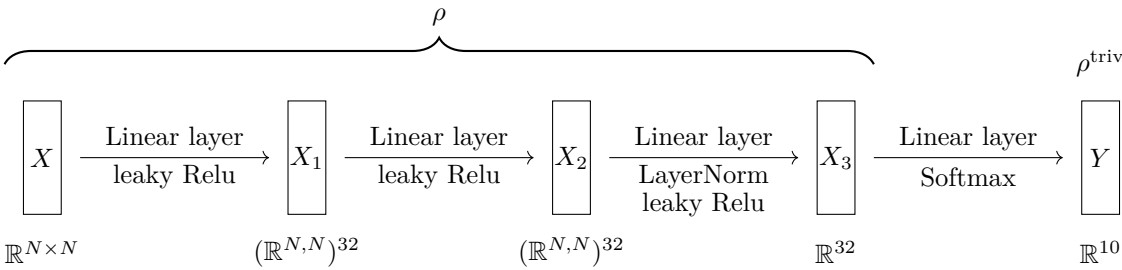

Figure 6: The nominal architecture for the second set of experiments. Note that the symmetry group differ between experiments.

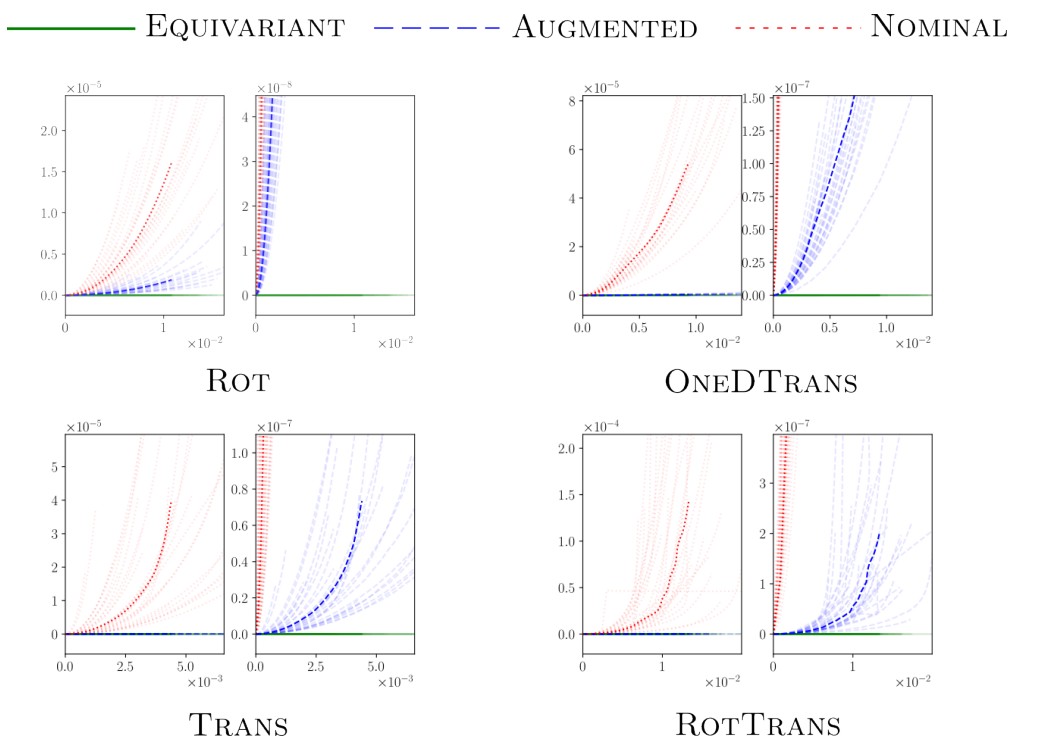

Figure 7: Results from the experiments of the appendix.

- The $x_{\text{left}}$-limit in both subplots is chosen as 1.5 times the maximal (with respect to the 50 training epochs) median (with respect to the 30 runs) value of $\|A - A_0\|$ for the EQUI model.

- The $y_{\text{left}}$-limit in the left subplot is chosen as 1.5 times the maximal (with respect to the 50 training epochs) median (with respect to the 30 runs) value of $\|\Pi_{\mathcal{E}^\perp} A\|$ for the NOM model.

- The $y_{\text{right}}$-limit in the left subplot is given by $\lambda \cdot y_{\text{left}}$, where $\lambda > 0$ is a factor common for all four groups. $\lambda$ is chosen so that $y_{\text{right}}$ is equal to 1.5 times the maximal (with respect to the 50 training epochs) median (with respect to the 30 runs) value of $\|\Pi_{\mathcal{E}^\perp} A\|$ for the AUG model *for the TRANS experiment.*

In this way, we ensure that the coordinate systems are on the same scale relative to the NOM and EQUI models. It is hence not surprising that the NOM curves look the same in all the plots – it is very much that way by design. The same is however not true for the AUG curve, which seems to depend heavily on the symmetry group used.

To exactly explain this dependence is beyond the scope of this work. However, a simple counting of the dimension of the spaces $T\mathcal{E}$ in each case seems to give some clue. In Table 2, we write down the dimensions of $\mathrm{Hom}_G(U, V)$ for the different $U$ and $V$ appearing in the experiments (the simple but tedious justifications of these numbers are given in the next subsection). Using these numbers, one calculates that the relative dimensions $T\mathcal{E}/T\mathcal{L}$ for the different groups approximately equal

$$
\begin{array}{llll}
\textsc{Trans}: & 5.1 \cdot 10^{-3} & \textsc{Rot}: & 0.25 \\
\textsc{OneDTrans}: & 7.1 \cdot 10^{-2} & \textsc{TransRot}: & 1.3 \cdot 10^{-3}
\end{array} \cdot \tag{17}
$$

We see a clear trend – the smaller the dimension of $T\mathcal{E}$, the the slower the $\textsc{Aug}$ architecture seems to deviate. This is not surprising – the closer $T\mathcal{E}$ is to $T\mathcal{L}$, the closer the augmented gradients and Hessians are to their non-augmented counterparts. To formally analyse this effect, in particular for more complicated $\mathcal{L}$ than considered here, is interesting future work.

### E.2.1 The spaces $\mathrm{Hom}_G(U, V)$ for the various groups

Here, we derive descriptions of $\mathrm{Hom}_G(U, V)$ in the four cases. These derivations are all conceptually easy, but tedious, and are only included for completeness.

$\mathbb{Z}_N^2$ **and Trans** It is well known that a linear operator $\mathbb{R}^{N,N} \to \mathbb{R}^{N,N}$ is equivariant to circular translations (i.e. the $\textsc{Trans}$ -action) if and only if it is a convolution. Consequently, the dimension of the space $\mathrm{Hom}_{\mathbb{Z}_N^2}(\mathbb{R}^{N,N}, \mathbb{R}^{N,N})$ is $N$. As for $\mathrm{Hom}_{\mathbb{Z}_N^2}(\mathbb{R}^{N,N}, \mathbb{R})$, notice that we by duality might as well may describe all $X \in \mathbb{R}^{N,N}$ that are not changed by any translation – and those are clearly exactly the ones which are constant. Hence, $\dim(\mathrm{Hom}_{\mathbb{Z}_N^2}(\mathbb{R}^{N,N}, \mathbb{R})) = 1$.

$\mathbb{Z}_N$ **and OneDTrans** To describe the space $\mathrm{L}_{\mathbb{Z}_N}(\mathbb{R}^{N,N}, \mathbb{R}^{N,N})$, let us begin by introducing some notation. First, every element in $\mathbb{R}^{N,N}$ can equivalently be described as a collection of rows. This can be written compactly as

$$
X = \sum_{k \in [N]} e_k x_k^* \tag{18}
$$

where $x_k \in \mathbb{R}^N$ are the rows, and $e_k$ is the $k$:th canonical unit vector. Correspondingly, each operator $A: \mathbb{R}^{N,N} \to \mathbb{R}^{N,N}$ decomposes into an array of $N^2$ operators $A_{\ell,k}: \mathbb{R}^N \to \mathbb{R}^N$:

$$
A(x) = \sum_{k,\ell \in [N]} e_\ell (A_{\ell,k} x_k)^*. \tag{19}
$$

With this notation introduced, we can conveniently describe the space $\mathrm{L}_{\mathbb{Z}_N}(\mathbb{R}^{N,N}, \mathbb{R}^{N,N})$

| Experiment | $U$ | $V$ | $\dim \mathrm{Hom}(U,V)$ | $\dim \mathrm{Hom}_G(U,V)$ | # in model |
|---|---|---|---|---|---|
| $\textsc{Rot}$ | $\mathbb{R}^{N,N}$ | $\mathbb{R}^{N,N}$ | $N^4$ | $\frac{1}{4}N^4$ | $32 + 32 \cdot 32$ |
| | $\mathbb{R}^{N,N}$ | $\mathbb{R}$ | $N^2$ | $\frac{1}{4}N^2$ | $32 \cdot 32$ |
| | $\mathbb{R}$ | $\mathbb{R}$ | $1$ | $1$ | $32 \cdot 10$ |
| $\textsc{Trans}$ | $\mathbb{R}^{N,N}$ | $\mathbb{R}^{N,N}$ | $N^4$ | $N^2$ | $32 + 32 \cdot 32$ |
| | $\mathbb{R}^{N,N}$ | $\mathbb{R}$ | $N^2$ | $1$ | $32 \cdot 32$ |
| | $\mathbb{R}$ | $\mathbb{R}$ | $1$ | $1$ | $32 \cdot 10$ |
| $\textsc{OneDTrans}$ | $\mathbb{R}^{N,N}$ | $\mathbb{R}^{N,N}$ | $N^4$ | $N^3$ | $32 + 32 \cdot 32$ |
| | $\mathbb{R}^{N,N}$ | $\mathbb{R}$ | $N^2$ | $N$ | $32 \cdot 32$ |
| | $\mathbb{R}$ | $\mathbb{R}$ | $1$ | $1$ | $32 \cdot 10$ |
| $\textsc{TransRot}$ | $\mathbb{R}^{N,N}$ | $\mathbb{R}^{N,N}$ | $N^4$ | $\frac{1}{4}N^2$ | $32 + 32 \cdot 32$ |
| | $\mathbb{R}^{N,N}$ | $\mathbb{R}$ | $N^2$ | $1$ | $32 + 32 \cdot 32$ |
| | $\mathbb{R}$ | $\mathbb{R}$ | $1$ | $1$ | $32 \cdot 10$ |

Table 2: Dimension calculations for the experiments $\textsc{Rot}$ , $\textsc{OneDTrans}$ and $\textsc{TransRot}$ in the appendix.

**Proposition E.1.** *Using the notation* (19), $\mathrm{L}_{\mathbb{Z}_N}(\mathbb{R}^{N,N}, \mathbb{R}^{N,N})$ *is characterized as the set of operators for which each $A_{\ell,k}$ is convolutional. In particular, the dimension of the space is $N^2 \cdot N = N^3$.*

*Proof.* Somewhat abusing notation, let us denote the action of $\mathbb{Z}_N$ on $\mathbb{R}^N$ also as $\rho^{\mathrm{tr}_0}$. We then have

$$\rho^{\mathrm{tr}_0}(n)(e_k x^*) = e_k \rho^{\mathrm{tr}_0}(n)(x)^* \tag{20}$$

Hence,

$$A(\rho^{tr_0}(n)x) = \sum_{k,\ell\in[N]} e_\ell (A_{\ell,k}\rho^{\mathrm{tr}_0}(n)x_k)^* \tag{21}$$

$$\rho^{\mathrm{tr}_0(n)}(Ax) = \sum_{k,\ell\in[N]} e_\ell(\rho^{\mathrm{tr}_0}(n)A_{\ell,k}x_k)^* \tag{22}$$

These two expressions are equal for any $x$ if and only if $A_{\ell,k}\rho^{\mathrm{tr}_0}(n)x_k = \rho^{\mathrm{tr}_0}(n)A_{\ell,k}x_k$ for any $x_k$ and $\ell, k$, i.e., when every $A_{\ell,k}$ is translation equivariant, which is equivalent to each $A_{\ell,k}$ being a convolution operator. $\square$

As before, it is not hard to realize what the space $\mathrm{L}_{\mathbb{Z}^N}(\mathbb{R}^{N,N}, 1)$ must be: Interpreted as a matrix $X = \sum_{k\in[N]} e_k x_k^*$, $X$ is in $\mathrm{L}_{\mathbb{Z}^N}(\mathbb{R}^{N,N}, 1)$ if and only if each $x_k$ is constant. Correspondingly, the projection is given by taking means along the $x$-direction, and the dimension of the space in particular is $N$.

$\mathbb{Z}_4$ **and Rot** A map $K$ in $\mathrm{Hom}(\mathbb{R}^{N,N}, \mathbb{R}^{N,N})$ can be described by an array $A[i,j]$ of real numbers, $i, j \in [N]^2$. The lifted representation $\overline{\rho}^{\mathrm{rot}}$ acts on this matrix as follows:

$$(\overline{\rho}^{\mathrm{rot}}(k)A)[i,j] = A[\omega^k i, \omega^k j], \quad k \in \mathbb{Z}_4.$$

This makes it clear that a map is in $\mathrm{Hom}_{\mathbb{Z}_4}(\mathbb{R}^{N,N}, \mathbb{R}^{N,N})$ if an only if $A$ is constant on each orbit $\{(\omega^k i, \omega^k j) \mid i, j \in [N]^2, k \in \mathbb{Z}_4\}$. When $N$ is even, all such orbits contain 4 elements, which implies that $\dim \mathrm{Hom}_{\mathbb{Z}_4}(\mathbb{R}^{N,N}, \mathbb{R}^{N,N}) = \dim \mathrm{Hom}(\mathbb{R}^{N,N}, \mathbb{R}^{N,N})/4 = N^4/4$. We may apply exactly the same argument to argue that $\mathrm{Hom}_{\mathbb{Z}_4}(\mathbb{R}^{N,N}, \mathbb{R}) = \dim \mathrm{Hom}(\mathbb{R}^{N,N}, \mathbb{R})/4 = N^2/4$.

$\mathbb{Z}_N^2 \rtimes \mathbb{Z}_4$ **and TransRot** It is clear that an element is invariant to the action $\mathbb{Z}_N^2 \rtimes \mathbb{Z}_4$, if and only if it is invariant to the subgroups $\mathbb{Z}_N^2$ and $\mathbb{Z}_4$. This immediately implies that, using the notation of Example 3.6

$$\mathrm{Hom}_{\mathbb{Z}_N^2 \rtimes \mathbb{Z}_4}(\mathbb{R}^{N,N}, \mathbb{R}^{N,N}) \subseteq \mathrm{Hom}_{\mathbb{Z}_N^2}(\mathbb{R}^{N,N}, \mathbb{R}^{N,N}) = \mathcal{C}$$

$$\mathrm{Hom}_{\mathbb{Z}_N^2 \rtimes \mathbb{Z}_4}(\mathbb{R}^{N,N}, \mathbb{R}) \subseteq \mathrm{Hom}_{\mathbb{Z}_N^2}(\mathbb{R}^{N,N}, \mathbb{R}) = \mathrm{span}\,\mathbb{1},$$

i.e. that linear maps must be convolutions, and that linear forms must be constant. The constant form is clearly rotation invariant, so that $\mathrm{Hom}_{\mathbb{Z}_N^2 \rtimes \mathbb{Z}_4}(\mathbb{R}^{N,N}, \mathbb{R})$ indeed equals $\mathrm{span}\,\mathbb{1}$. As for the rotation invariant convolutions, we now from the discussion in Example 3.6 that the rotation acts directly on the convolutional filter. This filter is hence constant if and only if it is constant on orbits of the rotation group, of which there (for even $N$) exists exactly $N^2/4$. Hence, $\mathrm{Hom}_{\mathbb{Z}_N^2 \rtimes \mathbb{Z}_4}(\mathbb{R}^{N,N}, \mathbb{R}^{N,N}) = N^2/4$, as claimed.

