# OpenReview forum: "Optimization Dynamics of Equivariant and Augmented Neural Networks"
_TMLR — Accepted by TMLR_

### Review · Reviewer_Wu6p · 2024-09-24

**Summary Of Contributions:**

This work presents a detailed analysis and comparison of the training dynamics between a model where equivariance is enforced by constraining its linear layers and a model where equivariance is promoted through data augmentations. The authors provide theoretical proofs for three main results regarding the training dynamics of these two models using gradient flow. Specifically, they show that:
- First, under some natural assumptions, the two optimizations share the same stationary points in the space of equivariant models.
- Second, the stable stationary points (in the space of equivariant models) of the model trained using only data augmentations are also stable stationary points for the equivariant model, but not vice versa.
- Third, under an additional assumption, the authors provide a disentanglement of a model trained with data augmentations into an equivariant and a non-equivariant part and determine the training dynamics of each part.

To emphasize the significance of these theoretical results, the authors also provide examples of practical scenarios where the corresponding assumptions hold, along with simple experimental results that support their theoretical observations.

**Audience:**

Yes

**Broader Impact Concerns:**

There are no concerns on the ethical implications of the work that would require adding a Broader Impact Statement.

**Claims And Evidence:**

Yes

**Requested Changes:**

Major (more critical changes)
- Related to the first weakness, the authors should  more explicitly state whether the assumption of the existence of a normalized Haar measure constraints the group $G$ for which the theoretical results hold.

Minor (changes that may strengthen the work)
- Related to the second weakness, a more detailed discussion regarding Assumption 2 and the cases where it becomes more challenging to satisfy would strengthen the completeness of this work.
- While this work does not consider the stochasticity of SGD, including experimental results for different batch sizes could provide a better understanding of how the batch size and the number of augmentations per batch affect the observed results.

Typos:
- At the end of Section 3.1 the reference is missing or is incorrect.
- In the proof of Proposition 3.12 (line 6):  should $\langle \dot{A},W\rangle=\langle X,W\rangle$  be $\langle \dot{A},W\rangle=\langle \dot{X},W\rangle$ instead?

**Strengths And Weaknesses:**

Strengths:
- The authors formulate a general framework that allows them to analyze the training dynamics for a wide range of equivariant neural networks. They showcase the generality of their formulation by providing detailed descriptions of how currently popular architecture can be reduced to their proposed framework.
- The authors provide a detailed analysis of the relationship between the training dynamics of an equivariant model and a model trained using data augmentations. The theoretical results presented in this paper are an important step toward a better understanding of the effects of data augmentation and their relation to the training of equivariant models.
- The theoretical results are presented clearly in this paper. The provided examples and experimental results facilitate the understanding of the practical implications of the theoretical findings.

Weaknesses:
- One of the initial assumptions in this work is the existence of a normalized Haar measure. Does this assumption constrain the group $G$ for which the theoretical results hold?  For example, how does the requirement for normalization affect the results in the case of non-compact groups like the $\mathbb{R}^n$.
- In Section 3, Assumption 2 states that the nonlinearities used in both the equivariant model and the model trained with data augmentations should be equivariant. While this assumption trivially holds for point-wise non-linearities when the intermediate representations $\rho_i$ are scalar features, it is more limiting when the intermediate representations include non-scalar equivariant features e.g. vector features. In the latter case, the point-wise non-linearities commonly used in non-equivariant models that are trained with data augmentations will not be sufficient to satisfy the assumption.

---

> ### Author Response · Authors · 2024-10-09
>
> First, thank you for the review. It is insightful and constructive. Let us in the following comment on both the weaknesses and requested changes. We will upload an updated version of the article as soon as the additional experiments that we will make are ready.
>
> ## Weaknesses
>
> + Normalized Haar measure
>
> The reviewer is completely right that this means that we mean to assume that the group is compact. We should have stated this more clearly. The results are not applicable for non-compact groups -- then, it is however also hard to choose a group element 'uniformly' at random. The question of non-Haar-distributed augmentation is interesting in general, but it is beyond the scope of this work. We will reformulate the global assumptions to make this more clear.
>
> + Non-linearities
>
> Again, the reviewer is completely in the right. It was not our intention to say that the equivariance of the non-linearities *always* is there, but point out that it often is. It makes sense to point out that there are relevant cases where the assumption is not fulfilled. We have do so, and also add some discussion. Note if the non-linearities are not equivariant, the restriction strategy is not sound, in the sense that an $A \in \mathcal{E}$ will not imply that $\Phi_A$ is equivariant, so that a theoretical comparison is hard in this case. One can of course still ask the question of the effect of augmentation in this case, but we deem it beyond the scope of this article.
>
> We have also added a few references to examples where more complicated equivariant non-linearities are constructed in the literature.
>
> ## Requested changes
> As discussed above, we will both point out the compact group assumption, as well as discuss non-trivial equivariant non-linearities. We will also perform new runs, both for smaller batches as well as one run of 'true' gradient descent, of the main set of experiments of the paper. We will report on the results of these as soon as they are ready. Finally, thanks for catching the typos, which we will correct.

---

### Review · Reviewer_xzLB · 2024-09-25

**Summary Of Contributions:**

The paper investigates and compares two strategies for optimizing neural networks when there is symmetry in data — equivariant architectures and data augmentation. The authors prove that under certain compatibility conditions, both strategies lead to the same equivariant stationary points. However, some stationary points that are stable for equivariant models may be unstable in augmented models. Numerical experiments on convolutional networks show that under unitary parametrization of layers, the equivariant space is invariant under the augmented flow.

**Audience:**

Yes

**Claims And Evidence:**

Yes

**Requested Changes:**

The theoretical results are strong and contribute meaningfully to understanding the optimization of neural networks on symmetric data. However, the paper would benefit from additional analysis on whether the theoretical results hold, even approximately, when using gradient descent instead of gradient flows, as well as more discussions of the practical relevance of the results.

**Strengths And Weaknesses:**

Strengths
- The results on stationary points and their stability in networks trained with data augmentation are novel and represent a significant contribution for understanding the dynamics of optimizing tasks with data symmetry.
- The paper is very well written and easy to follow. The mathematical framework for neural network is simple but general enough to cover many popular architectures, both with and without bias.
- The assumptions are clearly stated, explained, and justified.
- The experiments, though simple, effectively illustrates theoretical results (Theorem 3.7). The comparison between cross-shaped and skew-shaped filters in convolutional neural networks effectively shows that under unitary parametrization of layers, the equivariant space is invariant under the augmented flow.

Weaknesses
- The theory is based on gradient flow, whereas the experiments use stochastic gradient descent. Since the authors attribute some observations in experiments to this gap, more discussion on how the theoretical analysis extends to gradient descent would be beneficial.
- Although the authors state that Assumption 1 (that the group acts through unitary representations) is not a true restriction, it appears to restrict the analysis to compact groups. Also, it may be helpful to include the definition of unitary representation in the paper.
- There is little discussion on the practical relevance of the theoretical results. More concrete takeaways for practitioners, such as guidance on architecture design or how to choose between equivariant models and data augmentation, would be valuable.

---

> ### Author Response · Authors · 2024-10-09
>
> First, let us thank for a constructive and insightful review. Let us in the following comment on both the weaknesses and requested changes. We will upload an updated version of the article as soon as the additional experiments that we will make are ready.
>
> ## Weaknesses
>
> * Gradient flow vs. stochastic gradient descent.
>
> This comment got us thinking, and it made us realize three things: (1) Our first main result, about the *stationarity* of points, directly carries over to gradient descent - the stationary points of a gradient descent and the stationary point of the gradient flow are the same. (2) The same cannot be said about the second main result - if one uses a too big learning rate, points that are stable under the gradient flow will not be under gradient descent. In the low learning rate limit, however, the result should also carry over. (3) For the *stochastic* gradient descent, more work is simply needed, e.g. because the notion of stationarity change meaning. The stability results probably have some relevance, but here, both the size of the learning rate and how dependent the risk is on the random draw of batches plays a role. To theoretically quantify this will take considerable work, and we deem it beyond the scope of this paper. We will incorporate this discussion in the final version of the paper by extending Remark 3.8 and adding one after Theorem 3.11.
>
> * Compactness of the group
>
> As the comments of Reviewer xzLB made us realize, we indeed need to assume that $G$ is compact globally -- else, the Haar measure of the group will not be finite. We will clarify this in an updated version.
>
>
> * Practical discussion
>
> This is definitely a sound point to make. The issue is a bit subtle, because our results can both be used to argue for augmentation and for restriction of the architecture. First, supporters of either strategy can claim that the other one will find the same stationary points as ones own. The 'restricters' can argue that their strategy will make more points stable -- but the 'augmenters' can on the other hand argue that that induces more local minima on E, that we could 'escape' through allowing ourselves to take a detour outside $\mathcal{E}$. The'restricters' can then again argue that we are not guaranteed to come back to E after such an excursion. In short, our results do not give a clear, definite answer to the question whether augmentation or restrictions is better. We will add this discussion to the conclusion.
>
> ## Requested changes
> First, refer to the discussion of gradient flow vs. gradient descent above. In addition, we will perform a new set of runs for the experiments in the main paper, where we both test to lower the batch size of the SGD, as well as one run of non-stochastical gradient descent. This will hopefully give some more intuition about the practical relevance of our results. We will report on the results of these as soon as they are ready.

---

### Review · Reviewer_JkFh · 2024-10-07

**Summary Of Contributions:**

The submission investigates the optimization dynamics of neural network trained with augmentations and with constrained parameter space. In particular, it investigates the dynamics of the gradient flow and the nature of stationary points for the two training strategies. Under mild conditions it proves that stationary points, that lie in the affine subspace of admissible (equivariant) maps, are the same as those found with an unconstrained architecture by augmenting (by integrating over the group with the Haar measure) the dataset. Interestingly, a stationary point, in the affine subspace of admissible maps, can be stable for the constrained architecture and unstable for unconstrained architecture trained with augmentation, but not the other way around. Last, depending on the parameterization of the constrained architecture, namely the embedding map into the larger linear space, the set of stationary points can be invariant under the gradient flow of the augmented architecture.

**Audience:**

Yes

**Broader Impact Concerns:**

There aren't any concerns on ethical implications I can think of, since this paper is a theoretical analysis of existing methods.

**Claims And Evidence:**

Yes

**Requested Changes:**

There are minor suggested changes, apart from those suggested above, and typos:

- A double "that" in the abstract.
- Section 2, second paragraph, $\rho_Y$ should be $\rho_V$.
- Section 2.2, the space $\mathcal{E}$ is used but not yet defined.
- Section 3.1, last paragraph, it should be $\bar{\rho}(k)\mathcal{L}$, twice. And there is a missing reference in the end.
- In the proof of lemma 3.1, in the induction step, at the end, $A$ is missing an index $i$.
- Section 4.1, last paragraph there are two "when".

**Strengths And Weaknesses:**

Apart from minor issues discussed below the submission is well written and the notation is consistent throughout the manuscript. Despite this not being my main area of expertise I could follow the proofs and did not find any issues with the derivations. The results are certainly of interest to the community given the relevance of the long standing "inductive bias" vs. augmentation problem in ML. I cannot guarantee that these results are entirely novel because I am not very familiar with the relevant literature but to the best of my knowledge the optimization dynamics of constrained and augmented neural networks have never been studied like so and the proofs are novel.

There are some minor issues with the writing.
- The action of a representation on the $\mathcal{H}_G$ that is a direct product of subspaces. I guess the direct sum representation would be block diagonal?
- The first part of section 3 is a bit unclear. In particular, I don't fully understand the role of the base point $A_0$ and how you go from the system in $c$ to the one in $A$ by applying the chain rule. I am probably misunderstanding something but perhaps a few extra lines would help.
- Can you show one example of compact $G$ where the representations can be made unitary by redefining the inner product?
- In the experiment section you mention a random embedding. Can you elaborate? Maybe explicitly write what this embedding looks like.


Overall, I find the submission worth of being accepted because of clarity of the narrative and because it shows potentially interesting results for the community.

---

> ### Author Response · Authors · 2024-10-09
>
> First, thank you for the review. We are in particular happy to hear that our manuscript was understandable for the reviewer, although they did not consider themselves as an expert on the topic. The reviewer still has some concerns, that we will address in the following.
>
> ## Weaknesses
>
> + The action of the group on $\mathcal{H}_G$
>
> Indeed, the representation is defined like this shortly before Proposition 3.4. Note that by definition, the representation $\overline{\rho}$ on $\mathcal{H}_G$ is trivial.
>
> +  Beginning of Section 3
>
> As for the basepoint, note that any affine subspace can be written as the sum of one point in the affine subspace plus a vector subspace. We choose this point equal for both $\mathcal{L}$ and $\mathcal{E}$ to ensure that $c=0$ corresponds to the same point in both  parametrizations.
>
> As for the chain rule argument, let us first say that we will clearly have $\dot{A}=L \dot{c}$. The chain rule now implies, for instance, $\nabla \mathrm{R}(c) = L^*\nabla R(A_0 + Lc) = L^*\nabla R(A)$. Putting those two equations together,and $\dot{c}=-\nabla \mathrm{R}(c)$ yields the formula for the nominal dynamics. We have added this argument to the text.
>
> + Compact $G$ and unitary representations.
>
> This is a classical construction (see for instance [these lecture notes](https://www.math.toronto.edu/murnaghan/courses/mat445/ch6.pdf)). Since the inner product depends on the representation, giving a concrete example is almost as convoluted as describing the general case: Given a non-unitary representation $\rho$ of a group $G$ on a space $\mathcal{H}$ with inner product $\langle{\cdot, \cdot}\rangle$, we define a new inner product as so: $\langle{u,v}\rangle_G = \int_G \langle{\rho(g)u,\rho(g)v}\rangle d\mu(g)$, where $\mu$ is the Haar measure of the group. This is well defined -- the integral always converges, since $G$ is compact. One can also quickly show that $\rho$ is unitary with respect to the new inner product, via the invariance of the Haar measure: $\langle{\rho(h)u,\rho(h)v}\rangle_G = \int_G \langle{\rho(gh)u,\rho(gh)v}\rangle d\mu(g) = \int_G \langle{\rho(g)u,\rho(g)v}\rangle d\mu(g) = \langle{u,v}\rangle_G$. The finite-dimensionality is needed to show that the new inner product defines an *equivalent norm* to the one defined by the first one, which in essence means that the new inner product yields the same topology on the space $\mathcal{H}$.
>
> We are happy to include this argument in an appendix(it would not fit in the main text) if the reviewer so wishes.
>
> + Random embedding
>
> What we mean by a 'Gaussian distribution among all possible embeddings' is the following: As basis vectors for the space of filters, we draw random filters with cross-shaped support, with i.i.d Gaussian entries. We have clarified this in the text.
>
> ## Requested changes
>
> Thank you for the very careful proofreading. We will correct all these typos.

---

### Decision · Action_Editor_H4cV · 2024-12-07

**Recommendation:** Accept as is

**Comment:**

The authors make a commendable effort to analyze theoretically the relation between equivariance and data augmentation with respect to optimization. Under 'natural' assumptions on the data, they find that there is a direct relation, which I think is an interesting finding. As also the authors suggested, the experiments are not as detailed, which however is not unexpected given the theoretical focus of the work.

**Audience:**

Yes, the paper is certainly of interest to the TMLR audience, especially those interested in geometric deep learning. Optimization is an important aspect of training constrained neural networks, and this works provides theoretical insights in this direction.

**Claims And Evidence:**

Yes, as also reviewers confirm.